# A Study of the Impact of River Improvement and Greening on Public Reassurance and the Urban Well-Being Index during the COVID-19 Pandemic

**DOI:** 10.3390/ijerph19073958

**Published:** 2022-03-26

**Authors:** Hsiao-Hsien Lin, I.-Yun Chen, Chih-Hung Tseng, Yueh-Shiu Lee, Jao-Chuan Lin

**Affiliations:** 1School of Physical Education, Jiaying University, Meizhou 514015, China; chrishome12001@yahoo.com.tw; 2Department of Leisure Industry Management, National Chin-Yi University of Technology, Taichung 41170, Taiwan; boy217010@hotmail.com; 3Department of Finance, National Changhua University of Education, Changhua 50074, Taiwan; yun_0303@hotmail.com; 4General Education Center, National Penghu University of Science and Technology, Penghu 880011, Taiwan; leeys@gms.npu.edu.tw; 5Department of Marine Leisure Management, National Kaohsiung University of Science and Technology, Kaohsiung 811213, Taiwan

**Keywords:** experience value, leisure involvement, leisure satisfaction, well-being, circular economy

## Abstract

This study aims to investigate the effect of river improvement and greening projects on people and the urban happiness index. First, the quantitative method was adopted, and data collected from 734 questionnaires were analyzed using SPSS 26.0 software. Then, the qualitative method was used, and semi-structured interviews were conducted to collect the opinions of 12 interviewees, including scholars, government employees, citizens, and practitioners. Finally, we discuss multiple comparison analysis testing. The survey results indicate that river improvement and greening projects could be conducted to take advantage of diverse ecological environments, urban transportation planning, and geographical location. Such projects can help people relieve stress, even during the COVID-19 pandemic; improve their physical and mental health; and enhance their environmental awareness. However, due to poor traffic flow, a low space utilization rate, and inflexible management practices, visiting these environments has posed a risk of infection. As a result, most respondents indicated that the leisure benefits of green fields are limited and not helpful for improving their physical and mental health or having fun. Additionally, survey responses by people from different backgrounds (*p* < 0.01) varied. Therefore, we believe that by providing a safe living environment, strengthening disaster prevention skills and cooperation against epidemics, reducing accident risks, improving leisure safety and fluency, and planning diverse leisure activities, we can improve people’s perception of environmental experiences, promote leisure participation, improve leisure satisfaction, and enhance well-being.

## 1. Introduction

Water resources breed all kinds of ecology, and freshwater resources are one of the basic energy sources for humankind’s survival and development [1]. There are different types of rivers [2], and some of them have a large basin and stable water volume [3], coupled with diverse ecosystems, fresh air, and a hospitable environment. These rivers are one of the important resources for humankind’s survival and development.

However, water volume and flow velocity in rivers vary according to the climate. When the water volume of the river is stable, clean, and hygienic and the ecological resources are abundant, it will improve people’s quality of life, meet the needs of urban construction [4], and assist in the move towards stable development [5]. Conversely, when the water quantity of the river is unstable, floods often develop, and the water is not clean. At this time, it threatens the safety of the people, destroys the environment, and damages the structures of villages [6]. In addition, people have been threatened by viral infection and death during the COVID-19 pandemic, which has caused people to feel panic, reduced their sense of security [6,7], and affected their physical and mental health [7,8]. All increase people’s anxiety and lead to a decline in the quality of life and a sense of distrust in the living environment [9,10]. Therefore, stabilizing river water resources and providing a safe living and leisure environment is of great significance to maintaining people’s physical and mental health and realizing a happy society and city.

Water conservancy projects aim to control and protect surface and groundwater resources [11], and they are also one of the means for humans to use river water resources stably [12]. Originally, human beings used them to address the problems of rivers polluted or damaged by natural elements or artificial development [13]. However, later studies have confirmed that although water conservancy projects can effectively solve the problems with river water volume and water quality [14], they may also cause irreversible damage to the local ecology [15,16]. Therefore, it has become the recent trend of water conservancy projects to build a development model in which humankind and natural ecology coexist and achieve the goals of sustainable water and circular economy [17,18].

Water sustainability is aimed at ensuring that people can have access to fresh and clean water in response to climate change, the global population surge, and increased freshwater shortages [19,20,21]. The current trend of water conservancy policy [22,23] is to embody concepts of water resource cycle and development models [21] based on the concept of the green economy, avoid excessive land reclamation, reduce river course changes, maintain the original appearance of the river as much as possible, use the existing river sediment and spacing of embankments to maintain the function of flood control and dredging [24], make good use of the land along the river, pay attention to the quality of the ecological environment, develop the area into a space for leisure and tourism activities, create recyclable and sustainable resources [25], and achieve the ideal of sustainable development of water resources.

The Daiyun Mountains in Fujian Province belong to the temperate and subtropical climate zones: rainy all year round, with an average temperature of 22.1 °C and precipitation of 1610.2 mm. The Mulan River has more than 360 tributaries, with a length of 105 km, an average water flow of 1.552 billion tons, and a drainage area of 1732 square kilometers [26]. A 74.5 km section of the Mulan River flows through the Licheng District and the Chengxiang District of Putian City, bringing abundant fresh water to the surrounding villages and residents [27]. However, the Mulan River has been prone to floods since ancient times, bringing major disasters every ten years and minor disasters every year [28,29,30]. In October 1999, a strong typhoon caused floods in the Mulan River. Nearly 60,000 houses in Putian City collapsed; 450,000 mu of farmland were flooded; 30,000 people were evacuated; and 20,000 students were forced to stop going to school temporarily [29]. It was obvious that the flood caused serious damage to the surrounding villages and environment of the Mulan River.

In the light of this, the Chinese government promoted a 20-year river water conservancy project. The silt dug from the river course channel was dried, treated and used for building an embankment. Then, the “soft mattress” was pressed into the embankment foundation and filled with drainage material to improve the drainage efficiency of the embankment building and reduce the slippage of the foundation soil [31]. This can aid in avoiding the subsidence of the embankment to stabilize and enhance the erosion resistance of the subgrade [27]. A total area of 5.5245 hectares of green space along the river from the Mulan Impounding Reservoir to the Mulan River Bridge was developed for activity spaces and various industries, including leisure, housing, vacation, experience, history and culture, tourism, and the green energy industry [32] to attract investment, create employment [27], and boost the economy. This area has at least doubled its population to more than 900,000 people compared to its population during the 1999 floods [32]. It is obvious that the government carrying out the projects has provided a safe living, leisure, and tourism environment, and the surrounding villages and people have benefited from it. This has been helpful for people to participate in leisure and tourism activities, has stimulated them to participate [33], and has improved their physical and mental health and quality of life so as to achieve the goal of building a happy city [34]. The current status of Mulanxi green space planning is shown in Figure 1.

A happy city is an ideal, a concept. It means that all citizens enjoy their jobs and live happy lives, and it refers to a society wherein the values of truth, goodness, and beauty are fully realized [35]. On the other hand, scholars believe that it is necessary to create a healthy, happy, safe, and harmonious living environment so that people can feel happy [36], and a happy city can be realized [37,38].

Water conservancy projects have been carried out to improve the Mulan River, reduce flood damage, develop green space, boost the local economy, and stimulate people to participate in leisure activities [39]. This may help improve people’s physical and mental health and quality of life and achieve the goal of building a happy city [40]. However, usually there is a gap between the expected effect and the real one of policy decisions [41] because the completion of projects usually brings about positive and negative results [37,38,39]. Therefore, scholars believe that different means and methods should be used to obtain the opinions of the stakeholders so as to figure out the answers helpful to understanding the effectiveness of policy decisions [42,43].

In addition, recent studies on rivers and water conservancy projects have been carried out based on river velocity, water volume, and water quality to analyze flood and ecological impacts [40,44,45,46,47]. In response to climate change, some researchers have begun to explore the utilization of water resources in reservoirs, rivers, and cities [30] and the impact of water conservancy projects on urban or rural development [42,48] and people’s lives [49,50]. Although there have been studies on people’s environmental experience value of [50,51] and satisfaction [52] with river and water conservancy projects, studies on issues such as leisure participation combined with well-being have not been carried out. Moreover, there has been no study on water conservancy projects, green space, and people’s environmental experience and happiness around the Mulan River. Therefore, the researchers believe that the research gap can be filled by analyzing the influence of the Mulan River water conservancy and greening projects on people and urban development from the perspective of environmental experience value, leisure participation, satisfaction, and well-being.

In conclusion, the scope of this study is the Mulan River, local people are the subject, and it is designed to analyze the people’s environmental experience value, leisure involvement, and satisfaction and well-being with the green space created by the water conservancy project in the hope of verifying whether the river improvement greening can provide local residents with a safe and comfortable leisure environment space, promote personal physical and mental health, bring happiness, and establish a happy life from the people’s perspective. Hopefully, this study will provide new directions for future policy making and research, which is the purpose and focus of this study.

## 2. Literature Discussions

### 2.1. Well-Being

Happiness is a psychological attitude that emerges from positive experiences and is also a complex and subjective concept [35]. Well-being refers to an individual’s current emotions and preferences, and its measurement is based on personal subjective feelings [53]. Well-being is achieved when the material needs of life are satisfied and one has a sense of security in the atmosphere of the surrounding living environment [54].

Scholars have argued that economic and other material indices, such as gross domestic product, should be used to measure well-being [55], but other scholars have stated that the inclusion of health status, welfare status, and other social development impact indicators [56] can fully encompass the happiness index. Well-being can be measured as subjective well-being, psychological well-being, or social well-being [57]. It can also be determined by comparing the material needs and expectations of personal life [52]. Additionally, researchers can analyze the interaction between people and their living environment from the perspective of personal experience by examining factors such as life satisfaction, physical and mental health, and self-affirmation [54]. Leisure behavior, social interaction, emotions, health, physical fitness, self-confidence, and sense of accomplishment have been extensively researched [57], and studies have indicated that people with different life backgrounds and education levels have different perceptions of satisfaction based on their needs [58]. Therefore, everyone has their own definition and understanding of happiness.

Therefore, based on the literature, we will examine the aspects of life satisfaction, physical and mental health, and self-affirmation in terms of leisure behavior, sociability, emotion, health, physical fitness, self-confidence, and fulfillment. The research themes will then be approached from the perspective of different contexts and leisure participation styles to gain insight into the case.

### 2.2. The Experience Value and Well-Being of Green Space

Green space along the riverbanks refers to the space of existing river banks used or reconstructed for planting or spaces and facilities that are beneficial to the public for leisure, entertainment, sports, etc. The environmental experience with green space along the riverbanks refers to the physical and mental satisfaction of participants with their surroundings [59]. Usually, there is motivation before the desired behavior is generated. After comparing with past experience, new sensations are generated in physiology and psychology, and they become part of personal perceived value [60], and then experience value can be formed.

The environmental experience value is the participants’ evaluation of local building facilities; leisure and recreational activities; and spaces after they experience the performance of facilities, products, or services [61]. Scholars believe that the evaluation of environmental experience value can be divided into the evaluation of logical value, emotional value, and practical value [62]. It can be explored from the aspects of service superiority, consumer investment returns, fun, and aesthetics [63], and judged from the perspectives of education, escapism and aesthetics [64]. Among them, issues such as design style, integrity, attractiveness, promotion of conservation issues, environmental knowledge, and relaxation will be explored [65]. Moreover, studies have pointed out that people with different backgrounds have different feelings after experiencing the same environment or facility [66], and the higher the experience value felt, the better the positive psychological feelings of the people and the higher the well-being [67].

Therefore, based on the literature, we analyze and discussed the issues of design style, integrity, attractiveness, promotion of conservation issues, environmental knowledge, and relaxation in terms of education, escapism, and aesthetics. Then, the research topic is addressed from the perspectives of different contexts and leisure participation styles in order to gain insight into the case.

### 2.3. Leisure Involvement and Well-Being

Involvement is a psychological experience, as well as a perception [68]. Leisure involvement refers to a psychological state in which individuals are affected by internal and external factors, such as their inner core values, continuity, and physical environment or products, services, and communication, which affect their decision process of participation or purchase. These factors help people fulfill the valuable plan [69].

Leisure involvement refers to the continual involvement and participation in a certain thing that the individual feels is important mainly due to personal needs, interests, and viewpoints [70]. The higher the degree of leisure involvement, the deeper the self-involvement and the better the attention focus on leisure is [71]. The better the leisure behavior and performance, the happier one can feel [72], which is helpful for personal physical, and mental health, thus producing a feeling of happiness [71,72]. Scholars thus believe that leisure involvement can be divided into behavioral involvement and social involvement [73]; these dimensions can be defined based on people’s behaviors and attitudes. Leisure involvement can also be analyzed in terms of attractiveness, centrality, and self-expression, and it can also be understood according to how involved a person feels in a given setting. An individual’s participation and feelings, values, degree of affection, self-expression, self-confidence, and attention-seeking behavior are other areas of research [72,73]. Leisure activities can be examined in terms of how they enhance self-confidence and physical fitness, restore physical and mental health and physical strength, and increase knowledge and social interaction; additionally, self-exploration, relaxation, environmental experience, and feelings are worth investigating.

Therefore, based on the literature, we examined the issues of design style, integrity, attractiveness, enhancement of conservation issues, environmental knowledge, and relaxation in terms of attractiveness, centrality, and self-expression. Then, the research topic was addressed from the perspective of different contexts and leisure participation styles in order to gain insight into the case.

### 2.4. Leisure Satisfaction and Well-Being

Leisure satisfaction is the positive perception that an individual feels after he/she is aware of his/her leisure experience and environment when he/she engages in leisure activities [74]. It can also be said that leisure satisfaction is evaluated according to a person’s degree of satisfaction obtained from leisure experience.

Leisure satisfaction is mainly based on personal activity experience compared to previous experience, personal expectations, or sense of expectation [74]. When the actual leisure environment or activity content meets the individual’s expectations, a satisfied attitude is generated in his/her mind [75]. Scholars believe that the higher a person’s feelings of leisure satisfaction, the higher the positive energy cognition is obtained [76] and the more personal satisfaction can be obtained. The person is more satisfied with their current life and feelings, and personal well-being is further improved [75,76]. Scholars believe that the factors affecting leisure satisfaction can be classified as internal and external factors [77]. They can be analyzed according to six aspects: physiology, psychology, education, aesthetics, social interaction, and relaxation [78]. They can also be analyzed from the perspectives of relaxation, education, psychology, aesthetics, society, etc. [79,80]. Among them, more in-depth exploration can be conducted into the content of leisure activities for enhancing self-confidence and physical fitness, restoring physical and mental health and physical strength, increasing knowledge and social interaction, self-exploration and relaxation, environmental experience and feelings, etc. Moreover, studies have pointed out that people with different backgrounds and leisure modes would be significantly different from each other in the perception of leisure satisfaction [81].

Therefore, based on the literature, we explored participation and feelings, likes and values, expressing oneself, building self-confidence, and attracting attention through theories of leisure participation in terms of attractiveness, centrality, and self-expression. Then, the research topic was addressed from the perspective of different contexts and leisure participation styles in order to gain insight into the case.

## 3. Methods

### 3.1. Research Framework and Hypotheses

First, the researchers reviewed the literature to understand the impact of rivers on humankind’s development [1,2,3,4,5,6,7,8,9,10,11,12,13,14,15,16,17,18,19,20,21,22,23,24,25] and then further understand the development and status of related research [26,27,28,29,30,31,32,33,34,35,36,37,38,39,40]. Research studies have found that although the 10 km long reclaimed land has been developed into 5.5245 hectares of green spaces and 16 leisure and tourism activity spaces and scenic spots to create facilities and spaces for leisure, housing, vacation, experience, history and culture, characteristic tourism, and the green energy industry, there is still a gap between the expected effect of policy decisions and the actual situation, which usually occurs after the end of policy-making promotion [36,37,38,39,40,41]. Using different research approaches and methods and exploring the issues from the participant’s perspective can produce real answers [53,54,55,56,57]. Therefore, this study is designed to adopt a mixed research method to collect residents’ opinions and analyze whether the current greening measures of rivers and water conservancy projects are effective. It is helpful to understand the influence of river improvement and greening projects on humankind and urban development. The research framework is shown in Figure 2.

Researchers believe that the Mulan River had flooded quite regularly, and the local residents could not stand the hardship [28,29,30]. After the river water conservancy projects completed by the Chinese government, not only has the flood risk decreased, but 5.5245 hectares of green space have been created for living, leisure activities, and economic development [27,32]. It is apparent that the public should appreciate the environment and facilities improvement policy because the environment and facilities are helpful for people to engage in leisure activities, gain good experience, and improve their health and well-being [34,67,72,73,79]. However, there will always be a gap between the expected goal of policy making and reality, and it must be ascertained by those who experience it personally [42,43]. Based on the above literature and framework diagram, seven hypotheses and arguments have been proposed, described as follows.

**Hypothesis** **1** **(H1).**
*It is assumed that people’s cognition of environmental experience value is inconsistent.*


**Hypothesis** **2** **(H2).**
*It is assumed that people’s cognition of leisure involvement is inconsistent.*


**Hypothesis** **3** **(H3).**
*It is assumed that people’s cognition of leisure satisfaction is inconsistent.*


**Hypothesis** **4** **(H4).**
*It is assumed that people’s cognition of well-being is inconsistent.*


**Hypothesis** **5** **(H5).**
*It is assumed that environmental experience value and well-being cognition have a positive and significant influence.*


**Hypothesis** **6** **(H6).**
*It is assumed that leisure involvement and well-being cognition have a positive and significant impact.*


**Hypothesis** **7** **(H7).**
*It is assumed that leisure satisfaction and well-being cognition have a positive and significant influence.*


### 3.2. Research Process

As there is no relevant literature discussion on the current research topic, the theoretical foundation is relatively weak, so sufficient theoretical support is needed. First, the quantitative research method was used to supplement the research breadth [68], and then the qualitative research method was used to increase the depth of the research [69], which can atone for the shortcomings of research method or theory [68]. Therefore, for this study, first, the relevant literature was referred to [37,38,39,40,41,42,43,44,45,46,47,48,49,50,51,52,53,54,55,56,57,58,59,60,61,62,63,64,65,66,67,68,69,70,71,72,73,74,75,76,77,78,79,80], the questionnaire tool was edited, and then three experts were invited to check the content validity. After the amendment, 100 questionnaires were collected in June 2021, analyzed using SPSS 26.0 statistical software, and confirmed as the official questionnaire tool. Then, 900 questionnaires were distributed from July to September of the same year, with 743 valid questionnaires returned, and the recovery rate was 82.56%. After analyzing the questionnaire data using basic statistics, a *t*-test, an ANOVA test, and a PPMCC test, the semi-structured interview method was used to collect the opinions of officials, experts, and residents based on the results of the questionnaire analysis. Next, all the data were collated in a rigorous, sequential, and logical manner, and then valuable information was concluded through summarizing, organizing, and sorting methods [72,73,74,75,76,77,78,79,80,81,82,83,84]. Finally, multiple comparison analysis testing was used to conduct the discussion in a multi-data and multi-perspective manner [83,84,85,86,87].

### 3.3. Research Tools Design and Analysis

The mixed method was adopted to conduct the research, with the quantitative method used first. The questionnaire was edited based on the literature related to envi-ronmental experience value [59,60,61,62,63,64,65,66,67], leisure involvement [68,69,70,71,72,73], leisure satisfaction [74,75,76,77,78,79,80,81], and well-being [53,54,55,56,57]. The questionnaire is divided into three parts. The first part is the background of the respondents, including gender (male, female); age group (under 20, 21–30, 31–40, 41–50, 51–60, 61 and above); and education attainment (below junior high school, high school or vocational high school, junior college or university, graduate school or above). The second part is the leisure mode. There are leisure participation types (water activities, hiking, traveling, running, cycling, other activities); participation frequency (daily, five times a week, four times a week, three times a week, and two times a week); and so on. The third part is the variables, divided into 47 questions on environmental experience value, leisure involvement, leisure satisfaction, and well-being. It was designed with a Likert 5-point scale (1 point means very dissatisfied, 5 points mean very satisfied). After the questionnaire was edited, scholars in the fields of water resources, public policy and development, tourism planning, ecology, and environmental engineering were invited to validate the questionnaire. Six interviewees, including a local village official, a local senior citizen, an ordinary citizen, a tourist, and a leisure company operator, were also invited to express their opinions on the results of the subsequent data analysis. The relevant background and the interview issues are shown in Table 1.

After determining the issue, SPSS 22.0 statistical software was used to carry out statistical tests. The Kaiser–Meyer–Olkin (KMO) test is a statistical measure mainly to confirm the relationship between variables; the more common factors the variables have, the more suitable they are for factor analysis [86,88]. When the test result KMO > 0.6 and the *p* value in the Bartlett test is less than 0.01 (*p* < 0.01), it indicates that the scale is suitable for continuous factor analysis [89]. If α is greater than 0.60, the issue with high reliability [90] can be analyzed further.

Based on the literature [59,60,61,62,63,64,65,66,67], six questions about environment experience value cognition were designed after the editing. The analysis result showed that KMO was 0.849, while Bartlett’s approximate χ^2^ value was 317.007, df was 15, and the significance was *p* < 0.001, suitable for factor analysis. The explainable variations of the scale were 38.18%, 27.592%, and 19.293%, and the total explained variation was 85.065%. After factor analysis, good and reliable issues were retained. Then, they were named the issues of education (two questions), escapism (two questions), and aesthetics (two questions). The α coefficient was 0.929–0.948, and the total scale α coefficient was 0.949. Based on the above analysis results, it can be determined that this questionnaire has high reliability.

Based on the related literature [68,69,70,71,72,73], there were 13 questions about leisure involvement after the editing. The analysis result showed that KMO was 0.864, while Bartlett’s approximate χ^2^ value was 1368.454, df was 105, and the significance was *p* < 0.001, suitable for factor analysis. The explainable variations of the scale were 41.355%, 37.186%, and 10.042%, and the total explained variation was 88.582%. After factor analysis, good and reliable topics were retained. They were named attractiveness (five questions), centrality (four questions) and self-expression (four questions). The α coefficient was 0.981–0.983, and the total scale α coefficient was 0.984. Based on the above analysis results, it can be determined that this questionnaire has high reliability.

Based on the related literature [74,75,76,77,78,79,80,81], there were 21 questions about leisure satisfaction after the editing. The K analysis results showed that MO was 0.882, while Bartlett’s approximate χ^2^ value was 2273.5, df was 276, and the significance was *p* < 0.001, suitable for factor analysis. The explainable variations of the scale were 43.254%, 26.636%, 15.723%, 2.269%, 2.133%, and 1.344%, and the total explained variation was 91.359%. After factor analysis, good and reliable issues were retained. They were named physiology (four questions), psychology (three questions), education (three questions), social (four questions), relaxation (three questions), aesthetics (four questions), and other factors. The α coefficient was 0.987–0.988. The total scale α coefficient was 0.988. Based on the above analysis results, it can be determined that this questionnaire has high reliability.

Based on related literature [51,52,53,54,55], there were seven questions about well-being after the editing. The analysis results showed that KMO was 0.882, while Bartlett’s approximate χ^2^ value was 2273.5, df was 276, and the significance was *p* < 0.001, suitable for factor analysis. The explainable variations of the scale were 44.095%, 27.974%, and 14.918%, and the total explained variation was 86.987%. After factor analysis, good and reliable issues were retained. They were named as life satisfaction (two questions), physical and mental health (two questions), and self-affirmation (three questions). The α coefficient was 0.964–0.966, and the total scale α coefficient was 0.969. Based on the above analysis results, it can be known that this questionnaire has high reliability.

### 3.4. Scope and Object

Although the leisure environment and facilities created by the Mulan River improvement and greening project are mainly located in the Licheng District and the Chengxiang District of Putian City, the surrounding cities have been also affected. In addition, because China’s water conservancy engineering technology was not advanced in the early days [28], and the climate has changed dramatically [30], the topography and soil structure along the Mulan River have been damaged, so the soil erosion and damage to the cities caused by the flood have worsened [28,89]. Therefore, the Chinese government has carried out the Mulan River improvement project [32] while greening the land on both sides of the river [25]. At the same time, it has solved the problems of surrounding cities, promoted people to engage in leisure activities and urban development [27], and attempted to achieve the goal of building happy cities [34]. Therefore, the local and surrounding villages have benefited from this, and the current population has more than doubled, reaching 923,000 [91,92], enjoying the benefits of and engaging in leisure, recreation, and tourism activities in the surrounding areas. It can be seen that the policy-making of greening measures for water conservancy projects has a certain influence on the local area and surrounding villages.

The surrounding villages have benefited from this, and their population has doubled to 923,000 [90,91]. Moreover, everyone enjoys the benefits brought by the river improvement project. It is obvious that the policy-making of greening measures for water conservancy projects has a certain influence on the local area and surrounding villages, and the researchers can gather the opinions of local residents to understand the actual effect of policy making.

### 3.5. Methods and Limitations

The mixed method was adopted. However, when the researchers conducted field surveys at the site, they simultaneously conducted questionnaire sampling using the convenience sampling method coupled with the assistance of the online questionnaire platform to collect data. However, due to the threat of the COVID-19 pandemic during the survey and the limitations of funding, workforce, and material resources, the survey and sampling process were hindered. Therefore, various methods, such as conceptual sampling and snowball sampling, were used in the later stage to speed up the progress of the questionnaire survey. In addition, six interviewees, including experts in the fields of decision analysis, tourism, leisure, environmental management, and local seniors, as well as operators, were invited. The researchers first used the video system and telephone to obtain the interviewees’ consent to the interview and then used a semi-structured interview method to conduct the survey, asking them to express their opinions on the analysis and results of the questionnaire. Finally, all the data were collected and analyzed using multiple comparison analysis testing.

However, as explained above, due to the risk of COVID-19 infection and research methods and restrictions when the survey was carried out, there may have been insufficient sample numbers or errors in the data during data sampling. These possible deficiencies will be mentioned in recommendations for future research in the hope that subsequent researchers will remedy them.

### 3.6. Ethical Considerations

In this paper, a mixed-methods survey was conducted to examine the environmental leisure and tourism environment created by the Mulan River rectification and greening project. The residents in the neighboring villages were the subjects. The interviewees were those who had an understanding of the research issues or had personal experience in participating in the Mulan River project. Others were experts, scholars, or people in professional fields such as policy-making analysis, tourism, leisure, and environmental management. Therefore, all respondents were aware of the research topic and understood the content of the questionnaire. Furthermore, they agreed to anonymously provide the data without compensation. The research design and data collection process complied with ethical standards, and no ethical certification was required [91,93].

## 4. Results and Analysis

### 4.1. Analysis of Sample Background and Leisure Mode

A total of 743 questionnaires was returned for this study, and narrative statistics were used to analyze the respondents’ background and leisure mode information. It was determined that 53% of them were male and 47% female; the majority of them (56.1%) had junior college or university degrees, and the minority (8.1%) had master’s degrees or above; and the largest age group was 21–30 (41.5%), with the smallest being over 51 (7.8%). As for their leisure activities, the largest proportion of them liked to go hiking (35.4%), and the smallest liked cycling (7%); the largest proportion of them engaged in leisure activities twice a week (29.8%), and the smallest five times a week (13.7%).

It can be seen that among the local people engaging in leisure activities, males aged between 21–30 with college or university degrees exercised twice a week and mostly took a walk or went hiking; the smallest proportion was females 51 and above, engaging in cycling.

### 4.2. Analysis of Cognitive Differences in Environmental Experience Value, Leisure Involvement, Leisure Satisfaction, and Well-Being

A *t*-test, an ANOVA test, and a PPMCC test were used to explore people’s environmental value experience, leisure involvement, leisure satisfaction, well-being perception, and difference analysis, as shown in Table 2 and Table 3.

#### 4.2.1. Environmental Experience Value

Review of the literature revealed that the environmental experience value is the participant’s evaluation of local building facilities, leisure and recreational activities, and spaces after they experience the performance of facilities, products, or services [61]. However, there may be errors in the effectiveness of policy development [37,38], and people with different backgrounds and leisure modes may have perception gaps in the environmental experience value [66,67], so the researcher believes that there should be differences in people’s perception of environmental value experience. Therefore, in this study narrative analysis, a *t*-test, an ANOVA test, and other methods were used to analyze whether there is a difference in people’s perception of environmental value experience. This is Hypothesis 1.

There were six questions about environmental experience value, and they were discussed based on education, escapism, and aesthetics. It was found that environmental issues (3.82), beautiful scenery (3.83), and overall planning (3.89) had the highest scores, while environmental knowledge (3.75), forgetting troubles (3.82), and design style (3.87) had the lowest scores. Then, the analyses were conducted based on the interviewees’ backgrounds, and it was found that there was no significant difference between genders, age groups, and leisure frequency groups for environmental experience value. However, there were significant differences between people with different educational levels and different types of leisure regarding environmental issues and beautiful scenery (*p* < 0.01). This shows that people with different backgrounds have different opinions, and multiple comparison analysis can be carried out. It was also found that those with graduate degrees and those who went walking for leisure gave more attention to environmental issues and beautiful scenery, respectively. This result is inconsistent with Hypothesis 1.

It is obvious that most people think that after the completion of the river improvement project, the reclaimed green space can be used for leisure activities, help people to recognize policy-making, and raise environmental awareness. However, the design style and the planning of the traffic flow in scenic spots have been poorly planned, so people cannot have positive experiences, benefit from leisure activities, or forget their troubles. Furthermore, only highly educated people agree that environmental awareness promotion is effective, and those who go walking for leisure think that they have a greater chance of enjoying the scenery.

#### 4.2.2. Leisure Involvement

Leisure involvement is a psychological state in which the individual feels a certain thing is important due to his/her needs and chooses to continue to be involved and participate [68,69]. However, there will be a gap between the expected and actual experience of policy development effects [67], so this study suggests that there should be differences in people’s cognition of leisure involvement. Therefore, in this study narrative analysis, a t-test and an ANOVA test were used to analyze whether there is a difference in people’s cognition of leisure involvement. This is Hypothesis 2.

There were 13 questions about leisure involvement explored based on attractiveness, centrality, and self-expression. It was found that feeling happy (3.89) and interested (3.89), focus on life (3.71), and attracting attention (3.87) had the highest scores, the activity being important (3.71); the high status in mind (3.63, and the reliance on facilities (3.60) had the lowest scores. Next, the analyses were conducted based on the different backgrounds. It was found that there were significant differences between age groups, people with different education levels, people who took part in different types of leisure, and people with different frequencies of leisure for feeling satisfaction, enjoyment, an important role, a good inner state, discussion of activities, reliance on facilities, self-expression, natural behavior, and attracting attention (*p* < 0.01). This shows that people from different backgrounds have different views, and multiple comparison analysis can be carried out. It was found that people with a junior high school level of education had stronger feelings about the focus on life; people with graduate degree had stronger feelings of reliance on facilities; people going walking as a leisure activity had stronger feelings about a good inner state; people engaging in activities four times a week had a stronger feeling of satisfaction; and people engaging in activities twice a week had stronger feelings about important roles, discussing activities, and expressing themselves. These results are inconsistent with Hypothesis 2.

It is obvious that going to the locale and engaging in leisure activities has become an essential part of many people’s daily life, not only attracting public attention but also bringing about feelings of happiness. However, there are other leisure environments and spaces to choose from, and this place is just one of those with environments and facilities for leisure activities. Only male respondents felt happy to be there. Those who exercise twice a week found that they can focus on life goals here and create topical interactions with friends by expressing themselves. Those who exercise four times a week feel satisfied with their leisure activities. People with graduate degrees pay more attention to facilities, and those with junior high school education regard this place as an important part of their daily life. People engaging in walking pay more attention to their personal exercise environment.

#### 4.2.3. Leisure Satisfaction

Leisure satisfaction means to use one’s personal experience to verify previous impressions, personal expectations, or sense of expectation [74], and the comparison result is regarded as the degree of satisfaction [76]. However, there will be a gap between the expected and actual experience of policy development effects [67], so this study suggests that there should be differences in people’s perception of leisure satisfaction. Therefore, in this study narrative analysis, a *t*-test and an ANOVA test were used to verify whether there were differences in people’s cognition of leisure satisfaction. This is Hypothesis 3.

There were 21 questions about leisure satisfaction, which are discussed from the aspects of physiology, psychology, education, social interaction, relaxation, and aesthetics. It was found that issues such as fondness for leisure (3.76), improving physical fitness (3.82), learning new things (3.82), attracting peers (3.83), stabilizing emotions (3.83), and good planning (3.89) have been more widely perceived. The importance of leisure (3.62), recovering physical strength (3.78), understanding oneself (3.66), a friendly leisure environment (3.77), doing the thing that makes one happiest (3.80), and cleanliness (3.75) have been less widely perceived. These results are inconsistent with Hypothesis 3.

Next, the analyses were conducted according to different backgrounds. It was found that when doing the things that make one happiest (3.82:3.77) males felt more happiness (*p* < 0.01). There were significant differences between age groups, people with different education levels, people engaging in different types of leisure activities, and people with different leisure frequencies for the importance of leisure, fondness for leisure, improving self-confidence, challenging oneself, a sense of achievement, improving physical fitness, learning new things, meeting friends, attracting peers, stabilizing emotions, doing the thing that makes one happiest, beautiful scenery, and good planning (*p* < 0.01). This shows that people with different backgrounds have different opinions, and multiple comparison analysis can be carried out. It was found that people with graduate degrees had stronger feelings about doing the thing that made them happiest, people who like jogging had stronger feelings about challenging themselves and paid more attention to leisure, and people who exercise twice a week had stronger feelings about stabilizing their emotions.

It can be seen that most people believe that although the maintenance and planning of the local environment can be improved, they can attract peers, learn new knowledge together, engage in leisure activities, improve physical and mental health, and have fun. However, at present, the leisure space and traffic flow are poorly planned, the sanitation is poor, people do not have enough leisure time for leisure activities, and it is impossible to for them to have fun and relax, so this area cannot become the main place for people to engage in leisure activities. Only those who exercise twice a week felt that doing exercise here helped their moods improve. People with graduate degrees felt happy. People who like jogging thought that jogging here was challenging.

#### 4.2.4. Well-Being

Well-being is a kind of cognition of the current emotions and preferences measured using personal subjective feelings [53]. When the material needs of personal life are satisfied and the atmosphere of the surrounding living environment is recognized [43], this is well-being. However, there will be a gap between the expected and actual experience of policy development effects [66,67], so the relevant research shows that there should be differences in people’s cognition of well-being. Therefore, in this study, the narrative analysis, a *t*-test, and an ANOVA test were used to analyze whether there is a difference in people’s perceptions of well-being. This is Hypothesis 4: that there is a difference in people’s cognition of well-being.

There were seven questions about well-being, which were discussed regarding the aspects of life satisfaction, physical and mental health, and self-affirmation. It was found that having life goals (3.96), being more optimistic about life (4.04), and social interaction (3.96) had the highest scores, while trying new things (3.94), feeling younger and more energetic (4.03), and liking the current leisure plan (3.92) had the lowest scores.

Next, the analyses were conducted according to different backgrounds. It was found that there were differences between the genders for optimism about life (4.00:4.08), acquiring different skills and abilities (3.95:3.90), and fondness for current leisure planning (3.85:3.99). Women perceived the first two items more clearly, and men perceived the last one more clearly. Then, there were no significant differences between age groups, people with different education levels, and people with different leisure activity preferences for optimism about life, acquiring different skills and abilities, fondness for current leisure planning, social interaction, having life goals, and feeling younger and more energetic (*p* < 0.01). This shows that people with different backgrounds have different opinions, and multiple comparison analysis can be carried out. It was also found that people in the age group of 31–40 perceived current leisure planning more clearly, people exercising twice a week perceived having life goals more clearly, and people exercising three times a week perceived feeling younger and more energetic more clearly. These results are inconsistent with Hypothesis 4.

It can be seen that most people believe that green spaces do promote social interaction, make people live more optimistically, and make personal aims in life clearer. However, there is no way to try new things and make people feel more youthful and energetic, so this is not the best place for leisure activities. Only people aged 31–40 agree with the benefits of recreational environmental planning. People exercising twice a week have life goals, while those exercising three times a week have improved their health.

### 4.3. Correlation Analysis of Environmental Experience Value, Leisure Involvement, Leisure Satisfaction, and Well-Being

Policy-making aims to solve local development obstacles and difficulties, improve the current economic and environmental situation of communities, and improve the quality of life of the peoples [39]. Scholars believe that perfect policy-making can withstand the test of time, gain favor and trust from the people, promote social harmony, create a high of quality life and a stable and harmonious living environment [24], enhance people’s happiness, and create happy cities [34]. Based on the above inferences, the researchers believe that the river rectification and greening project also aims to prevent flooding, promote local development, provide a leisure environment, and create a safe and happy city. Therefore, the local people’s environmental value experience, leisure involvement, leisure satisfaction, and happiness should also have a relevant influence. As a result, in this study the PPMCC test was used analyze and verify whether environmental value experience, leisure involvement, leisure satisfaction, and well-being have an influence on each other. These are Hypotheses 5, 6, and 7, respectively.

The analysis results demonstrate that environmental experience value, leisure involvement, satisfaction, and cognitive well-being had significant (*p* < 0.01) and positive effects; among them, leisure involvement (0.392) had the strongest influence, and environmental experience value (0.360) and leisure satisfaction (0.360) had the weakest influences. These results supported Hypotheses 5–7. Analysis results further indicated that attractiveness (0.412) and escapism (0.345) had a strong influence on the overall well-being and life satisfaction, physical and mental health, and self-affirmation of the respondents. Relaxation (0.358) had a strong influence on overall well-being, life satisfaction, and physical and mental health, whereas social interaction (0.354) had a strong influence on self-affirmation. All these results supported Hypotheses 5–7.

It can be seen that environmental experience value, leisure involvement, satisfaction and well-being have positive influences on each other. Among them, self-affirmation is most affected by attractiveness, escapism, and social interaction, while physical and mental health are most affected by attractiveness, escapism, and relaxation. Life satisfaction is most affected by attractiveness, escapism, and relaxation. The better and stronger the actual feelings of environmental experience value, leisure involvement, and satisfaction are, the higher the cognition of happiness is. As shown in Table 4 and Table 5.

### 4.4. Discussion

For this study, discussion on the respondents’ background and leisure model, environmental value experience, leisure involvement, leisure satisfaction, perception differences in well-being, and related issues will be carried out. Next, the effects and influence of leisure space created by river rectification and greening projects on the improvement of the happiness index of the people and the city will be discussed.

#### 4.4.1. Analysis of the Sample Background

The surroundings of the river are one of the main options for the development of human settlements. The scope discussed in this case is on both sides of the river. In addition to the river’s ecological and environmental advantages, different thematic environmental plans have been carried out in the later phase, and related public facilities have been built so that the local space for leisure, recreation and tourism activities is quite extensive. This is advantageous for the people who currently need a wide range of sports facilities, a large amount of activity, flexible time to plan and engage in various leisure sports, and a place to enjoy the river scenery at the same time. However, although the riverside is spacious, there are few shelters, plenty of sunshine, and too much space for leisure. In addition, pedestrian trails and bicycle lanes have not been separated clearly, and the middle-aged and elderly people have less physical strength. As a result, most of the respondents engaging in leisure activities are males, aged 21–30, who graduated from a junior college or university and go walking or hiking twice a week. As for female respondents, few of them are 51 or older, and they ride a bicycle for exercise.

#### 4.4.2. Regarding Environmental Experience Value

The researchers believe that placing soil in the river to reinforce the embankment on both sides of the river has reduced the development of existing river reservations. Additionally, numerous trees and shrubs have been planted to strengthen the soil structure. This has effectively reduced the damage to the soil structure of the river embankment in the event of heavy rain and has helped maintain the existing ecological environment. Furthermore, the air circulation along the river, the ecological diversity, and the spacious environment help disperse airborne pathogens, thereby reducing risk of disease transmission. The local government has created a flood control museum complex that combines the existing riverside ecology with the surrounding cultural and historical buildings; the museum is intended to promote political achievements and culture and to serve as a leisure and tourism site. Most respondents opined that the increase in green space from the river improvement has been beneficial and clearly demonstrated how policy can become reality; this is consistent with the literature [59,60].

However, respondents expressed displeasure with the similarity of the attractions, facilities, and leisure activities; moreover, they complained of too many people and how cyclist and pedestrian paths were not separate. Despite the pandemic ebbing and vaccination rates being high, people expressed concerns that vaccination efficacy is low and that crowding in green spaces undermines the intended low-risk atmosphere; if people are crowded, they feel their safety is jeopardized. This poor attitude toward green space leisure is inconsistent with the literature [61,62,63,64,65].

Furthermore, most people are busy with work and have limited leisure time. The bicycle operates like a vehicle, with speeds far greater than pedestrians, which can pose a serious hazard to both, so the environment needs to be improved for both of them. Only highly educated people agree that environmental awareness promotion is effective, and people who go walking for leisure think that they have a greater chance of enjoying the scenery. This result is inconsistent with the literature [66].

#### 4.4.3. Regarding Leisure Involvement

The researchers believe that the 10-km-long Mulan River green area has a diverse ecosystem with beautiful scenery. Flood controls, cultural and leisure spaces, and attractions dot the river bank, which is already a well-known location for leisure activities for locals. Additionally, the streets of surrounding villages are interlaced with bridges crossing the river, which enables convenient access for the public to engage in leisure activities. Therefore, for most locals, visiting the green area to engage in leisure activities has become an essential part of their everyday life. The park not only attracts public attention but also brings joy to the residents. This result is consistent with the literature [68,69].

Although the green area is spacious and contains various facilities, most of the facilities are simply open spaces for admiring the landscape and do not have staff in charge of management and maintenance. Additionally, visitors may have poor personal hygiene habits and insufficient epidemic prevention knowledge; failure to respect the personal space of others is a major problem in leisure activities and may dissuade some visitors from returning. Some survey respondents contended that the green space lacks compelling facilities/activities that would render it irreplaceable; thus, if they feel at risk, they will not come. This result is inconsistent with the literature [73,74].

The main purpose of leisure is to improve physical and mental health. Young people engage in leisure activities for stress relief and relaxation. Highly educated people typically experience pressure from work and social responsibilities, and, consequently, they expect to be able to exercise in a safe environment and improve their physical and mental health. Walking is one of the slowest-moving forms of exercise, and it may involve coming into contact or interacting with other people. Therefore, people who walk require a safe and healthy environment for exercise. Adequate exercise, at least twice a week, is the best means for improving physical fitness and mental health. Thus, respondents who exercised four times a week were the most satisfied, and those who exercised twice a week reported having a purpose in life, self-confidence, and topical interactions with friends. Respondents with a graduate degree paid more attention to facilities, whereas those with a junior high school education regarded this place as a vital personal space. People who exercise by walking paid more attention to their personal exercise environment. This result is inconsistent with the literature [73].

#### 4.4.4. Regarding Leisure Satisfaction

The researchers believe that after the Mulan River improvement project was completed, spaces have been provided for housing, leisure, and tourism activities and flood control for the cultural museum. In addition, the river flow is abundant, can help regulate the climate, and has a diverse ecosystem and beautiful environment. At present, the Mulan River has said goodbye to the negative image of flooding, and the green space by the river has become the main environment for most people to live and spend their leisure time. Therefore, most people believe that compared with the past, the environmental maintenance and planning of the green space has been improved and is helpful for people to engage in leisure activities with companions, learn new knowledge, improve physical and mental health, and, finally, have fun. This result is consistent with the literature [74].

However, although Chinese people have gradually resumed daily life due to the high COVID vaccination rate, not everyone has sufficient environmental knowledge; many people engaging in leisure activities do not like to wear masks, there still are many people littering and spitting in public places, and a safe and clean environment for leisure activities has not been provided. Therefore, most people believe that the current space and traffic flow have been poorly planned, the hygiene is poor, they cannot relax and have fun, and it cannot become a popular place for people to engage in leisure activities. This result is inconsistent with the literature [74,79,80].

Furthermore, the green space stretches for 10 km along the river; the area is vast; there are green facilities on both sides, including bicycles and jogging tracks; and it is helpful for people to engage in leisure activities such as jogging, especially for those men who like to do hard exercise. However, the risk for COVID-19 infection and people’s poor environmental knowledge has made the environment not safe enough for leisure activities. As a result, people who go jogging worry about the safety of the environment for sports, while people with insufficient knowledge of epidemic prevention and those who engage in leisure activities more frequently are prone to infection risks and cannot fully devote themselves to leisure activities, so their physical and mental health cannot be truly improved. Therefore, only men are happy. Those who engage in sports twice a week in the local area think that their mood can be lightened. Those with graduate degrees feel happy. People who go jogging think that doing exercise here is challenging. This result is inconsistent with the literature [81].

#### 4.4.5. Regarding Well-Being

The researchers believe that after the completion of Mulan River improvement and greening project, there are green spaces used for leisure activities and tourism, as well as cultural and recreational facilities. In addition, the streets in surrounding villages are interlaced with the bridges across the river, convenient for the public to go to engage in leisure activities. This is helpful for people to engage in leisure and tourism activities, improve their physical and mental health, relieve anxiety, and increase the effectiveness of their work and study practices. Therefore, most people believe that the green spaces do help promote social interaction, make them more optimistic about life, and make personal goals in life clearer. This result is inconsistent with the literature [53,54].

However, due to the small differences in the types of leisure and tourism facilities and the large differences in people’s awareness of epidemic prevention and environmental knowledge, people have doubts about the safety and hygienic quality of the leisure environment. In addition, there are other leisure, tourism or recreational facilities around the city, and there are many options. Long-term involvement in a leisure environment with risk of infection will not achieve the effect of physical and mental relaxation. Therefore, most people think that the green space is not the best place for leisure activities because they are not able to try new things and increase their vitality here. This result is inconsistent with the literature [35,52,54].

In addition, the area for leisure activities is large; the types of leisure that can be engaged in are highly selective. By engaging in leisure activities, the public can relieve stress, improve physical fitness, and increase the efficiency of work and study. However, for men, due to the differences in people’s epidemic prevention and hygiene knowledge, there is still a risk of infection in the environment, and they cannot fully devote themselves to leisure activities. Most of the facilities have been planned for tourism and do not meet the needs of young people. In addition, the environment is full of infection risks and traffic safety hazards. People over the age of 41 bringing young children to the place have safety concerns. It is also not conducive to the relaxation of middle-aged and elderly. Therefore, only women are optimistic about the environment, and men like the leisure planning here. People aged 31–40 favor the benefits of environmental planning for leisure activities. People exercising twice a week think they have clarified their purpose in life here, while those who exercise three times a week think they have improved their health. This result is inconsistent with the literature [58].

### 4.5. Relevance of Environmental Experience Value, Leisure Involvement, Leisure Satisfaction and Well-Being

The world is affected by the COVID-19, and society is still full of life, school, family, and work pressure. It has a certain impact on personal physical and mental health and quality of life. On the other hand, although the policy development will still have defects and cannot meet the needs of life, leisure, and tourism for all people, effective river rectification and greening projects can basically solve the flood problems, provide people with a safe and secure living environment, use the idle space on both sides of the river to plan and construct leisure facilities, help people to develop good leisure awareness and habits, and enhance physical and mental health and quality of life. In addition, the current plan has been able to meet the needs of the people in terms of leisure behavior, social interaction, emotion stabilization, health, physical fitness, self-confidence, and sense of accomplishment so as to create a good life and social environment, promote career development, and aid in achieving happiness goals. Furthermore, adequate leisure activities can relieve stress, have an effect of relaxation, and improve life satisfaction. Secondly, adequate leisure activities can be effective exercise, relieve stress, and improve physical and mental health. Finally, perfect leisure planning can increase self-confidence, expand social circles, provide more available resources, create more diversified career development opportunities, and aid in achieving personal goals for happiness.

Therefore, it is verified that there is a positive interaction between environmental experience value, leisure involvement, satisfaction, and happiness. Among them, self-affirmation is most significantly affected by attractiveness, escapism, and social interaction, while physical and mental health are most significantly affected by attractiveness, escapism, and relaxation. Life satisfaction is greatly affected by attractiveness, escapism, and relaxation. The better and stronger the actual feelings of environmental experience value, leisure involvement, and satisfaction are, the higher the perception of happiness is. This result is consistent with the literature [67,71,72,75,76].

### 4.6. Recommended Environmental Development Guidelines for Urban River Improvement and Greening

The right water engineering decisions can stabilize rivers, reduce flooding, increase green space, and provide adequate and healthy water resources and safe recreation and living environments, thus allowing cities to grow and people’s quality of life and well-being to improve. However, insufficient human resources for site management, the high similarity between the recreational environments, high pedestrian flow, poor recreational routes, poor public environmental literacy, poor environmental maintenance, and high environmental risks still affect the public’s recreational experience of rivers and green spaces and prevent the enhancement of the public’s sense of well-being.

Therefore, our recommendations are to strengthen the dike structure; increase green woodlands; provide a safe living environment; develop a recreational tourism environment by combining cultural, historical, and ecological resources; link evacuation and epidemic prevention information; strengthen disaster prevention skills and epidemic prevention cooperation; increase maintenance manpower and trash cans; maintain cleanliness; reduce the risk of accidents; divert the flow of people; plan recreational routes; enhance recreational safety and pedestrian flow; install recreational activity and facility managers; and plan diverse recreational activities. This will effectively improve the function and usage of the river and green space, build a safe living environment, and enhance people’s sense of well-being. The development trend concept is shown in Figure 3.

## 5. Conclusions

A well-developed water conservancy project can stabilize the amount of water in a river, reduce flooding, and increase green space. It also indirectly provides a safe recreational living environment, promotes urban development, and improves quality of life and happiness. However, the lack of human resources for on-site management, the high similarity between the recreational environments, the large number of people, and the poor recreational routes lead to a decrease in the experiential value. Lack of environmental knowledge, poor environmental maintenance, high environmental risk, and reduced recreational participation and satisfaction will lead to a lack of positive recreational experiences and well-being associated with green spaces.

Therefore, in this case, it is necessary to strengthen the embankment structure, increase green woodlands, provide a safe living environment, and develop a recreational tourism environment by combining cultural, historical, and ecological resources. In addition, the city should also provide evacuation and epidemic prevention information, strengthen disaster prevention skills and epidemic prevention cooperation, increase maintenance manpower and garbage bins, maintain cleanliness, reduce the risk of accidents, divert the flow of people, plan recreational routes to improve recreational safety and accessibility, assign managers of recreational activities and facilities, and plan diverse recreational activities. Doing so will effectively improve the function and usage of the river and green space, build a safe living environment, and enhance people’s sense of well-being.

Based on the above analysis results, the following suggestions are made for decision makers and managers, and enterprises, people, and future research directions.

1.For decision-makers and managers

It is recommended that the decision-making team include professionals required for policy goals who have influence in the field in the decision-making process; management needs to hire more a relevant workforce or combine with related industrial enterprises to provide sufficient facilities and activities planning so that the effectiveness of policy can be achieved.

2.For enterprises

It is recommended that nearby companies or individual studios use the available space and combine corporate resources and the workforce to promote leisure activities, which can create business opportunities for individuals or enterprises and increase local management workforce and scenic features.

3.For the public

It is recommended that the public make good use of park space, carefully choose activity time, encourage the planning of two to three leisure activities and then choose an adequate leisure activity based on personal athletic ability, time, and income, etc. so as to improve physical and mental health.

4.For future policy and research directions

It is recommended that the policy of river rectification and greening projects establish a sounder policy organization team, make good use of existing environmental space and resources, and plan reasonable and sustainable management methods. Secondly, due to the consideration of research limitations, the number of questionnaires, regions, and scope are limited. It is recommended that researchers of follow-up studies expand the scope of research, adopt different research methods, explore different issues, or use regional surveys for research that conducive to better planning and policies and achieving the goals of water recycling economic policy development.

## Figures and Tables

**Figure 1 ijerph-19-03958-f001:**
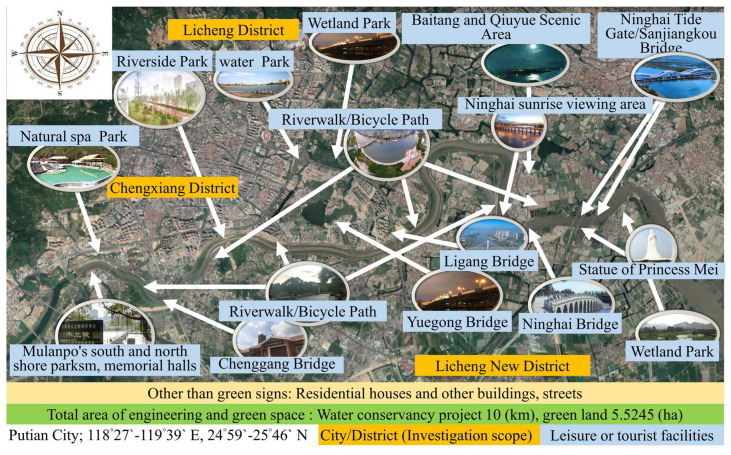
Distribution of the development and construction of the Mulan River improvement and greening project.

**Figure 2 ijerph-19-03958-f002:**
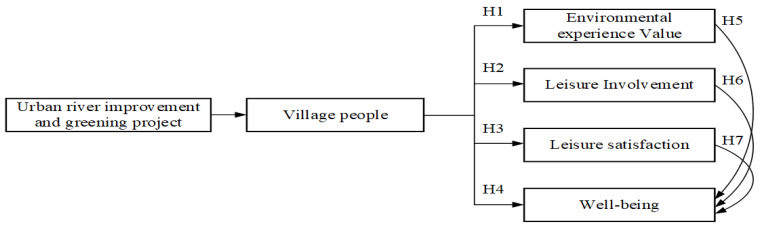
Study framework.

**Figure 3 ijerph-19-03958-f003:**
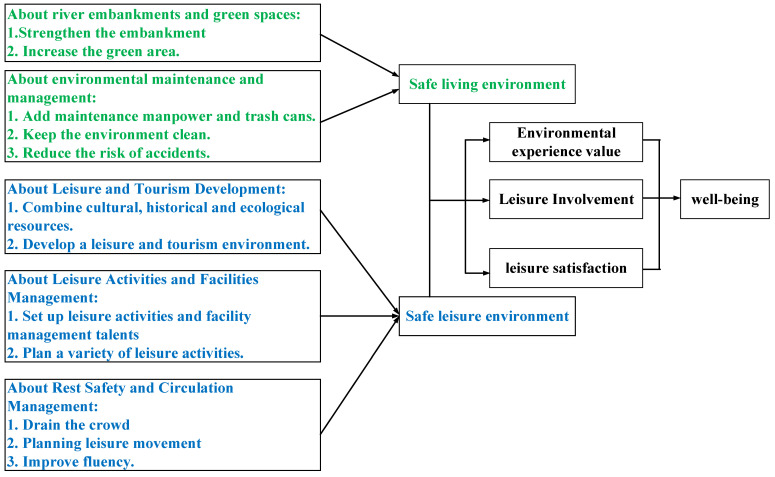
Developmental trend of urban river green space for leisure and recreation.

**Table 1 ijerph-19-03958-t001:** Respondents’ background information and an overview of the interview outline.

Identity/Research Areas	Gender	Residence Time/Years of Work Experience	Identity/Research Areas	Gender	Residence Time/Years of Work Experience
Resident (R1)	male	30	Water resources (P1)	male	12
Elder (El1)	male	60	Decision development (P2)	female	15
Elder (El2)	female	58	Leisure and sport (P3)	male	15
Tourist (T1)	female	40	Ecology (P4)	male	25
Entrepreneur (En1)	female	30	Environmental engineering (P5)	male	32
Entrepreneur (En2)	male	25	Village officer (V1)	male	20
Questionnaire	Issues				Inspectors
Environmental Experience Value	Understanding of environmental issues; Environmental knowledge; Beautiful scenery; Forgetting troubles; Design style; Improvement of overall planning.				P1; P2; P3; P4; P5
Leisure Involvement	Importance of activity; Satisfaction; Happiness; Interest; Fondness; Focus on life; Important role; Good inner state; Discussion of leisure activities; Reliance on facilities; To show oneself; Natural behavior; Attracting attention.				P1; P2; P3
Leisure Satisfaction	Importance of leisure; Enjoyment of leisure; Sense of accomplishment; Improved self-confidence; Challenge; Physical fitness; Rejuvenation; Broad vision; Learning new things; Understanding oneself; Making friends; Develop interpersonal relationships; Friendly leisure environment; Attracting companions; Feeling relaxed; Emotional stability; Doing the thing that makes one happiest; Environmental safety; Cleanliness; Beautiful scenery; Good planning.				P1; P2; P3
Well-being	Trying new things; Life goals; Feeliong younger and more energetic; Optimism about life.				P1; P2; P3; P4; P5
Questionnaire	Issues				Inspectors
Impact of tourism development	1. Please comment on the effectiveness of environmental planning of greening measures for river water conservancy projects.				P1; P2; P3; P4; P5; R1; El1; El2; T1; En1; En2; V1
	2. Please comment on the public utilization rate of the leisure space planned by the greening measures of the river water conservancy project.				
	3. Please comment on the effectiveness of recreational space planning planned by the greening measures of river water conservancy projects.				
	4. Please comment on the changes in the well-being of the local city after the implementation of the greening measures of the river water conservancy project.fo				

**Table 2 ijerph-19-03958-t002:** Analysis of cognitive differences in environmental experience value, leisure Involvement, leisure satisfaction, and well-being (environmental experience value and leisure involvement).

Factor	Subfactor	Issue	M	SD	Rank	Gender			Age							Education Level					
						G1	G2	*p*	Under 20	21–30	31–40	41–50	Over 51	*p*	Post Hoc	E1	E2	E3	E4	*p*	Post Hoc
Environmental Experience Value	Education	Understanding of environmental issues	3.82	1.14	1	3.82	3.83	0.781	3.67	3.86	3.92	3.87	3.69	0.348	N.A	3.72	3.87	3.75	4.28	0.006 *	graduate > junior, college
		Environmental knowledge	3.75	1.00	2	3.75	3.76	0.444	3.75	3.79	3.77	3.71	3.62	0.771	N.A	3.82	3.82	3.70	3.82	0.440	N.A
	Escapism	Beautiful scenery	3.83	1.10	1	3.82	3.84	0.682	3.77	3.76	3.99	3.99	3.72	0.066	N.A	3.76	3.93	3.74	4.22	0.007 *	graduate > college
		Forgetting troubles	3.82	1.04	2	3.84	3.83	0.532	3.70	3.80	4.08	3.92	3.69	0.014	N.A	3.62	3.93	3.82	3.95	0.114	N.A
	Aesthetic	Design style	3.87	1.06	2	3.88	3.86	0.273	3.76	3.81	4.11	3.96	3.78	0.038	N.A	3.62	3.93	3.82	3.95	0.457	N.A
		Improvement of overall planning	3.89	1.11	1	3.91	3.87	0.128	3.82	3.84	3.97	4.02	3.93	0.487	N.A	3.77	3.93	3.84	4.02	0.277	N.A
Leisure Involvement	Attractiveness	Importance of activity	3.71	1.135	4	3.74	3.67	0.597	3.76	3.72	3.56	3.71	3.78	0.526	N.A	3.81	3.96	3.85	4.10	0.064	N.A
		Satisfaction	3.86	1.091	2	3.86	3.86	0.382	4.01	3.86	3.62	3.83	4.02	0.072	N.A	3.50	3.86	3.70	3.55	0.103	N.A
		Happiness	3.89	1.024	1	3.94	3.84	0.177	3.97	3.85	3.88	3.79	4.16	0.046	N.A	3.64	3.99	3.83	3.92	0.132	N.A
		Interest	3.89	1.065	1	3.92	3.85	0.741	3.94	3.93	3.83	3.90	3.67	0.300	N.A	3.81	4.02	3.83	4.00	0.640	N.A
		Fondness	3.80	1.098	3	3.85	3.75	0.395	3.73	3.82	3.78	3.71	4.09	0.105	N.A	3.83	3.97	3.86	3.90	0.429	N.A
	Centrality	Focus on life	3.71	1.905	1	3.62	3.82	0.063	3.96	3.62	3.72	3.58	3.88	0.235	N.A	3.62	3.83	3.83	3.75	0.001 *	junior > senior
		Important role	3.70	1.229	2	3.76	3.65	0.740	3.73	3.72	3.58	3.69	3.90	0.599	N.A	3.23	4.14	3.62	3.70	0.098	N.A
		Good inner state	3.63	1.212	3	3.68	3.56	0.449	3.80	3.63	3.41	3.45	4.00	0.002 *	N.A	3.45	3.85	3.69	3.77	0.078	N.A
		Discussion of leisure activities topics	3.70	1.075	2	3.77	3.63	0.914	3.68	3.66	3.59	3.82	4.03	0.014	N.A	3.50	3.70	3.57	3.95	0.039	N.A
	Self-expression	Reliance on facilities	3.60	1.159	4	3.66	3.52	0.625	3.62	3.65	3.47	3.57	3.55	0.705	N.A	3.67	3.88	3.67	3.48	0.000 *	graduate > junior, senior
		To show oneself	3.72	1.087	3	3.79	3.65	0.221	3.77	3.70	3.72	3.67	3.81	0.898	N.A	3.19	3.78	3.55	3.85	0.253	N.A
		Natural behavior	3.86	1.062	2	3.87	3.86	0.827	4.06	3.76	3.67	3.98	4.07	0.005 *	N.A	3.51	3.81	3.72	3.73	0.082	N.A
		Attracting attention	3.87	3.188	1	3.62	4.15	0.058	3.63	3.68	3.60	4.96	3.79	0.314	N.A	3.74	4.03	3.83	3.73	0.000 *	N.A
Leisure Satisfaction	Physiological	Importance of leisure	3.62	1.184	4	3.58	3.67	0.363	3.56	3.63	3.70	3.76	3.29	0.160	N.A	5.31	3.80	3.68	3.60	0.008 *	N.A
		Enjoyment of leisure	3.76	1.115	1	3.75	3.78	0.900	3.82	3.70	3.72	3.96	3.67	0.112	N.A	3.60	3.69	3.53	4.07	0.209	N.A
		Sense of accomplishment	3.69	1.156	3	3.69	3.70	0.064	3.64	3.70	3.85	3.79	3.26	0.037	N.A	3.79	3.74	3.73	4.05	0.008	N.A
		Improved self-confidence	3.70	1.072	2	3.65	3.75	0.087	3.68	3.69	3.86	3.70	3.47	0.269	N.A	3.59	3.79	3.61	4.10	0.274	N.A
	Psychological	Challenge	3.80	1.105	2	3.89	3.71	0.813	3.77	3.73	4.02	3.87	3.71	0.093	N.A	3.73	3.73	3.64	3.92	0.028	N.A
		Physical fitness	3.82	1.039	1	3.78	3.86	0.344	3.80	3.73	3.91	4.08	3.64	0.004 *	N.A	3.74	3.92	3.72	4.10	0.001 *	N.A
		Rejuvenation	3.78	1.131	3	3.77	3.78	0.819	3.75	3.68	4.04	3.84	3.69	0.028	N.A	3.68	3.97	3.72	4.17	0.025	N.A
	Education	Broad vision	3.75	1.085	2	3.69	3.83	0.208	3.61	3.69	3.88	3.96	3.79	0.016	N.A	3.67	3.87	3.70	4.12	0.078	N.A
		Learning new things	3.82	1.137	1	3.82	3.83	0.781	3.67	3.86	3.92	3.87	3.69	0.348	N.A	3.69	3.84	3.69	4.03	0.006 *	junior > college, graduate
		Understanding oneself	3.66	1.178	3	3.71	3.61	0.852	3.51	3.67	3.84	3.71	3.52	0.171	N.A	3.72	3.87	3.75	4.28	0.016	N.A
	Social contact	Making friends	3.68	1.117	4	3.71	3.65	0.340	3.62	3.71	3.90	3.66	3.26	0.003 *	N.A	3.54	3.72	3.59	4.08	0.006 *	N.A
		Developing interpersonal relationships	3.80	1.050	2	3.82	3.78	0.590	3.60	3.80	3.92	3.94	3.78	0.082	N.A	3.59	3.69	3.62	4.15	0.013	N.A
		Friendly leisure environment	3.77	1.148	3	3.83	3.70	0.483	3.68	3.74	3.92	3.76	3.84	0.494	N.A	3.82	3.85	3.72	4.18	0.017	N.A
		Attracting companions	3.83	1.106	1	3.82	3.84	0.461	3.63	3.72	3.97	4.10	4.03	0.001 *	N.A	3.82	3.77	3.70	4.20	0.001 *	N.A
	Relaxation	Feeling relaxed	3.82	1.040	2	3.84	3.83	0.532	3.70	3.80	4.08	3.92	3.69	0.014	N.A	3.76	3.97	3.72	4.25	0.114	N.A
		Emotional stability	3.83	1.112	1	3.86	3.81	0.348	3.65	3.75	4.07	4.11	3.69	0.000 *	N.A	3.62	3.93	3.82	3.95	0.110	N.A
		The thing that makes one happiest	3.80	1.077	3	3.82	3.77	0.000 *	3.71	3.78	3.71	4.01	3.86	0.114	N.A	3.94	3.78	3.79	4.13	0.003 *	graduate > senior
	Aesthetics	Environmental safety	3.87	1.062	2	3.88	3.86	0.273	3.76	3.81	4.11	3.96	3.78	0.038	N.A	3.77	3.79	3.73	4.28	0.457	N.A
		Cleanliness	3.75	1.003	4	3.75	3.76	0.444	3.75	3.79	3.77	3.71	3.62	0.771	N.A	3.77	3.93	3.84	4.02	0.440	N.A
		Beautiful scenery	3.83	1.099	3	3.82	3.84	0.682	3.77	3.76	3.99	3.99	3.72	0.066	N.A	3.82	3.82	3.70	3.82	0.007 *	N.A
		Good planning	3.89	1.107	1	3.91	3.87	0.128	3.82	3.84	3.97	4.02	3.93	0.487	N.A	3.76	3.93	3.74	4.22	0.277	N.A
Well-being	Life satisfaction	Trying new things	3.94	1.092	2	3.90	3.99	0.012	4.04	3.88	4.04	3.83	4.02	0.339	N.A	3.81	3.96	3.85	4.10	0.116	N.A
		Life goals	3.96	1.093	1	3.93	3.99	0.884	3.98	3.91	4.08	3.93	3.98	0.666	N.A	3.92	3.99	3.96	3.62	0.210	N.A
	Physical and mental health	Feeling younger and more energetic	4.03	1.081	2	3.99	4.06	0.016	4.19	3.95	4.18	3.83	4.12	0.018	N.A	4.14	3.87	3.98	3.82	0.212	N.A
		Optimism about life	4.04	0.995	1	4.00	4.08	0.007 *	4.01	3.95	4.32	3.96	4.17	0.000 *	N.A	3.87	4.03	4.08	3.83	0.299	N.A
	Self-affirmation	Obtaining different skills and abilities	3.93	1.065	2	3.95	3.90	0.004 *	3.99	3.94	4.19	3.78	3.50	0.000 *	N.A	4.04	4.14	4.01	3.90	0.154	N.A
		Liking the current leisure plan	3.92	1.019	3	3.85	3.99	0.000 *	3.99	3.86	4.26	3.71	3.78	0.000 *	31–40 > 21–30, 41–50	4.04	4.04	4.04	4.04	0.654	N.A
		Social interaction	3.96	1.018	1	3.91	4.01	0.204	4.02	3.94	4.35	3.57	3.88	0.000 *	N.A	3.85	3.88	3.96	3.83	0.220	N.A

* *p* < 0.01. G1 (male), G2 (female); E1 (junior), E2 (senior), E3 (college), E4 (graduate).

**Table 3 ijerph-19-03958-t003:** Analysis of cognitive differences in environmental experience value, leisure involvement, leisure satisfaction, and well-being (leisure satisfaction and well-being).

Facet	Subfactor	Issue	Types of Leisure								Leisure Frequency						
			L1	L2	L3	L4	L5	L6	*p*	Post Hoc	F1	F2	F3	F4	F5	*p*	Post Hoc
Environmental Experience Value	Education	Understanding of environmental issues	3.70	3.68	3.91	4.14	4.00	3.80	0.007 *	walk > jog	4.01	3.81	3.51	3.93	3.84	0.006 *	N.A
		Environmental knowledge	3.63	3.69	3.73	4.05	3.86	4.20	0.012	N.A	3.92	3.81	3.76	3.47	3.70	0.008 *	N.A
	Escapism	Beautiful scenery	3.64	3.81	3.81	4.15	3.88	4.20	0.014	N.A	4.08	3.86	3.76	3.74	3.62	0.020	N.A
		Forgetting troubles	3.72	3.79	3.77	4.14	4.02	3.80	0.024	N.A	4.00	3.79	3.74	3.89	3.75	0.237	N.A
	Aesthetic	Design style	3.81	3.78	3.89	4.07	4.00	4.20	0.183	N.A	3.94	3.83	3.89	3.79	3.94	0.784	N.A
		Improvement of overall planning	3.72	3.81	3.92	4.21	4.02	4.20	0.010	N.A	4.07	3.86	3.76	3.86	3.92	0.247	N.A
Leisure Involvement	Attractiveness	Importance of activity	3.78	3.61	3.65	3.85	3.90	3.80	0.311	N.A	3.84	3.50	3.70	3.94	3.67	0.012	N.A
		Satisfaction	3.86	3.79	4.00	3.84	3.76	3.80	0.477	N.A	3.86	3.64	3.89	4.15	3.95	0.002 *	qiw > biw
		Happiness	3.87	3.82	3.94	3.99	4.00	3.80	0.649	N.A	3.98	3.73	3.96	4.06	3.80	0.050	N.A
		Interest	3.86	3.78	3.94	4.09	3.98	3.80	0.200	N.A	3.92	3.85	3.83	4.07	3.83	0.291	N.A
		Fondness	3.80	3.73	3.80	3.85	4.12	3.80	0.433	N.A	3.81	3.59	3.70	4.20	3.89	0.000 *	biw > qiw, tiw
	Centrality	Focus on life	3.66	3.51	4.06	3.74	3.67	3.80	0.108	N.A	3.77	3.47	3.71	4.02	3.77	0.219	N.A
		Important role	3.66	3.55	3.77	3.93	4.05	3.80	0.036	N.A	3.92	3.14	3.79	4.11	3.98	0.000 *	biw >qd, piw, qiw, tiw
		Good inner state	3.69	3.43	3.60	3.94	3.93	3.80	0.004 *	walk > jog	3.76	3.14	3.66	3.98	3.96	0.000 *	N.A
		Discussion of leisure activities topics	3.66	3.59	3.81	3.87	3.74	3.80	0.170	N.A	3.86	3.41	3.58	4.07	3.84	0.000 *	biw > qd, piw, qiw, tiw
	Self-expression	Reliance on facilities	3.66	3.44	3.73	3.69	3.55	3.80	0.118	N.A	3.54	3.23	3.75	3.91	3.88	0.000 *	N.A
		To show oneself	3.70	3.56	3.75	4.04	3.93	3.80	0.006 *	N.A	3.83	3.37	3.67	4.02	4.00	0.000 *	biw > qd, piw, qiw
		Natural behavior	3.82	3.70	3.95	4.13	4.05	3.80	0.009 *	N.A	4.02	3.62	4.00	4.08	3.70	0.000 *	N.A
		Attracting attention	3.86	3.45	4.61	3.67	3.93	3.80	0.012	N.A	3.66	3.27	4.93	4.11	3.73	0.000 *	N.A
Leisure Satisfaction	Physiological	Importance of leisure	3.41	3.49	3.74	3.96	3.76	3.80	0.002 *	jog > dabble, walk	3.88	3.30	3.51	3.80	3.83	0.000 *	N.A
		Enjoyment of leisure	3.62	3.67	3.79	4.11	3.93	3.80	0.009 *	N.A	4.04	3.57	3.70	3.70	3.90	0.002 *	N.A
		Sense of accomplishment	3.61	3.54	3.81	4.01	3.67	3.80	0.010 *	N.A	3.84	3.41	3.70	3.68	4.05	0.000 *	N.A
		Improved self-confidence	3.66	3.58	3.72	3.96	3.76	3.80	0.081	N.A	3.89	3.39	3.71	3.81	3.86	0.000 *	N.A
	Psychological	Challenge	3.76	3.65	3.68	4.31	4.17	3.80	0.000 *	jog > dabble, walk, trip	3.96	3.50	3.96	3.90	3.87	0.000 *	N.A
		Physical fitness	3.82	3.71	3.77	4.09	4.00	3.80	0.044	N.A	4.04	3.67	3.84	3.76	3.80	0.030	N.A
		Rejuvenation	3.71	3.63	3.77	4.16	3.98	3.80	0.003 *	N.A	3.94	3.63	3.72	3.86	3.78	0.161	N.A
	Education	Broad vision	3.66	3.77	3.69	3.91	3.86	3.80	0.504	N.A	3.96	3.71	3.59	3.67	3.86	0.059	N.A
		Learning new things	3.70	3.68	3.91	4.14	4.00	3.80	0.007 *	N.A	4.01	3.81	3.51	3.93	3.84	0.006 *	N.A
		Understanding oneself	3.59	3.51	3.72	3.95	3.83	3.80	0.029	N.A	3.92	3.50	3.66	3.75	3.50	0.016	N.A
	Social Contact	Making friends	3.75	3.50	3.75	3.89	3.67	3.80	0.036	N.A	3.71	3.62	3.58	3.71	3.83	0.568	N.A
		Developing interpersonal relationships	3.75	3.70	3.89	3.97	3.76	3.80	0.239	N.A	3.89	3.69	3.69	3.87	3.98	0.114	N.A
		Friendly leisure environment	3.70	3.58	3.76	4.27	4.00	3.80	0.000 *	N.A	4.01	3.73	3.73	3.74	3.56	0.052	N.A
		Attracting companions	3.84	3.78	3.73	4.09	3.93	3.80	0.150	N.A	3.98	3.71	3.79	3.95	3.75	0.179	N.A
	Relaxation	Feeling relaxed	3.72	3.79	3.77	4.14	4.02	3.80	0.024	N.A	4.00	3.79	3.74	3.89	3.75	0.237	N.A
		Emotional stability	3.68	3.76	3.80	4.22	4.00	3.80	0.003 *	N.A	4.12	3.65	3.84	3.80	3.78	0.005 *	biw > qd
		The thing that makes one happiest	3.76	3.75	3.76	3.97	3.95	3.80	0.487	N.A	3.90	3.62	3.84	3.82	3.91	0.127	N.A
	Aesthetics	Environmental safety	3.81	3.78	3.89	4.07	4.00	4.20	0.183	N.A	3.94	3.83	3.89	3.79	3.94	0.784	N.A
		Cleanliness	3.63	3.69	3.73	4.05	3.86	4.20	0.012	N.A	3.92	3.81	3.76	3.47	3.70	0.008 *	N.A
		Beautiful scenery	3.64	3.81	3.81	4.15	3.88	4.20	0.014	N.A	4.08	3.86	3.76	3.74	3.62	0.020	N.A
		Good planning	3.72	3.81	3.92	4.21	4.02	4.20	0.010 *	N.A	4.07	3.86	3.76	3.86	3.92	0.247	N.A
Well-being	Life Satisfaction	Trying new things	4.01	3.90	3.84	4.03	4.10	3.80	0.550	N.A	4.05	3.75	3.92	3.93	4.20	0.016	N.A
		Life goals	3.84	4.03	3.84	4.10	4.07	3.80	0.215	N.A	4.17	3.80	3.82	3.86	4.28	0.000 *	biw > piw
	Physical and Mental Health	Feeling younger and more energetic	3.99	4.04	4.01	4.11	3.98	3.80	0.926	N.A	4.29	3.92	3.76	4.03	4.21	0.000 *	tiw > qd
		Optimism about life	3.97	4.03	4.01	4.22	4.02	3.80	0.418	N.A	4.27	3.95	3.85	3.98	4.19	0.002 *	N.A
	Self-affirmation	Obtaining different skills and abilities	3.83	3.79	4.00	4.20	4.14	3.80	0.011	N.A	4.03	3.71	3.72	4.06	4.35	0.000 *	N.A
		Liking the current leisure plan	3.90	3.86	3.89	4.11	4.00	3.80	0.418	N.A	4.07	3.90	3.73	3.98	3.93	0.040	N.A
		Social interaction	3.99	3.90	3.93	4.07	4.00	3.80	0.778	N.A	4.18	3.91	3.58	3.83	4.40	0.000 *	N.A

* *p* < 0.01; L1 (dabble), L2 (walk), L3 (trip), L4 (jog), L5 (bicycle), L6 (other); F1 (qd), F2 (biw), F3 (tiw), F4 (qiw), F5 (piw).

**Table 4 ijerph-19-03958-t004:** Correlation analysis of environmental experience value, leisure involvement, leisure satisfaction, and well-being (environmental experience value and leisure involvement).

	Environmental Experience Value	Education	Escapism	Aesthetic	Leisure Involvement	Attractiveness	Centrality	Self-Expression
Well-being	0.360 **	0.342 **	0.345 **	0.329 **	0.392 **	0.412 **	0.348 **	0.308 **
Life satisfaction	0.289 **	0.271 **	0.286 **	0.259 **	0.317 **	0.327 **	0.303 **	0.234 **
Physical and mental health	0.341 **	0.322 **	0.328 **	0.311 **	0.358 **	0.394 **	0.311 **	0.271 **
Self-affirmation	0.359 **	0.346 **	0.336 **	0.332 **	0.400 **	0.412 **	0.344 **	0.332 **

** *p* < 0.01.

**Table 5 ijerph-19-03958-t005:** Correlation analysis of environmental experience value, leisure involvement, leisure satisfaction, and well-being (leisure satisfaction and well-being).

	Leisure Satisfaction	Physiological	Psychological	Education	Social Contact	Relaxation	Aesthetics
Well-being	0.360 **	0.308 **	0.340 **	0.353 **	0.338 **	0.358 **	0.333 **
Life satisfaction	0.286 **	0.242 **	0.269 **	0.272 **	0.270 **	0.292 **	0.270 **
Physical and mental health	0.323 **	0.256 **	0.316 **	0.314 **	0.300 **	0.329 **	0.314 **
Self-affirmation	0.374 **	0.336 **	0.348 **	0.376 **	0.354 **	0.361 **	0.330 **

** *p* < 0.01.

## Data Availability

Not applicable.

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
