# Peer review of "A Study of the Impact of River Improvement and Greening on Public Reassurance and the Urban Well-Being Index during the COVID-19 Pandemic"

_ijerph, 2022, doi:10.3390/ijerph19073958_

Round 1
Reviewer 1 Report
1. English writing of the whole manuscript is poor, and there exists too many grammatical errors. It needs to be rechecked and improved carefully. In addition, there are too many lengthy sentences which should be shortened.
2. The words of this manuscript are long-winded and boring, and some words are repeated too many times. The whole manuscript can be shortened a lot compared with its present content.
3. Many theoretical analysis or discussion aobut well-being should be closely related to hydraulic engineering. A lot of such contents are too common sense.
4. Environmental experience Value, Leisure Involvement, Leisure satisfaction,
5. How to judge the reasonability of the 7 hypothesis?
6. An obvious shortcoming in the investigation is that “the interviewees had an understanding of the research issues, had personal experience in participating, or were expert, scholar, and people in professional fields such as policy-making analysis, tourism, leisure, and environmental management”. This means that the interviewees could not represent all the residents.
7. In the investigation of environmental experience value, the cultural connotation in the design of the river rectification and greening project should also be considered.
8. All the conclusions are common sense in the present manuscript. It is suggested that quantitative findings or more valuable suggestions are given in the final part of this research.
9. Little detailed information is introduced about the real engineering project of the studied river. Please notice that the purpose of this research is to explore the influence of river rectification and greening projects on the well-being index of people and cities.
10. The title is not accurately describe the content of the manuscript. Where’s the study about environmental risk of epidemic?
Author Response
Reviewer 1
- English writing of the whole manuscript is poor, and there exists too many grammatical errors. It needs to be rechecked and improved carefully. In addition, there are too many lengthy sentences which should be shortened.
Dear reviewers:
Thanks for your suggestion. The manuscript has been sent to English professionals to assist with proofreading. Such as the red text of the manuscript. Hope to have your approval.
- The words of this manuscript are long-winded and boring, and some words are repeated too many times. The whole manuscript can be shortened a lot compared with its present content.
Dear Reviewers:
Thanks for your suggestion. We have rearranged and adjusted the entire manuscript. Such as the manuscript content in red. Hope to have your approval.
- Many theoretical analysis or discussion aobut well-being should be closely related to hydraulic engineering. A lot of such contents are too common sense.
Dear Reviewers:
Thanks for your suggestion.
With regard to some of the analyses and chapters, we have made substantial adjustments and clearly identified key factors. and discuss accordingly. as manuscript. Hope to get your approval.
- Environmental experience Value, Leisure Involvement, Leisure satisfaction,
Dear Reviewers:
Thanks for your suggestion.
With regard to some of the analyses and chapters, we have made substantial adjustments and clearly identified key factors. and discuss accordingly. as manuscript. Hope to get your approval.
- How to judge the reasonability of the 7 hypothesis?
Dear Reviewers:
Thank you for your suggestion. We are informed of possible expected outcomes based on discussions in the literature. And set research hypotheses according to its inference results.
When it is consistent with the inference, it means that the representative hypothesis is established, otherwise it is not established.
Brief description as in p. 6.
- An obvious shortcoming in the investigation is that “the interviewees had an understanding of the research issues, had personal experience in participating, or were expert, scholar, and people in professional fields such as policy-making analysis, tourism, leisure, and environmental management”. This means that the interviewees could not represent all the residents.
Dear Reviewers:
Thank you for your suggestion. We add in our manuscript the significance of the representativeness of the interviewees. As shown in Table 1. and the above description in red font.
- In the investigation of environmental experience value, the cultural connotation in the design of the river rectification and greening project should also be considered.
Dear Reviewers:
Thanks for your suggestion.
An investigation on the value of environmental experience. In the content of p.18-19, we have supplemented the cultural connotation in the design of river regulation and greening projects.
- All the conclusions are common sense in the present manuscript. It is suggested that quantitative findings or more valuable suggestions are given in the final part of this research.
Dear Reviewers:
With regard to some of the discussion sections, we have made significant adjustments and made clear the key factors.
- Little detailed information is introduced about the real engineering project of the studied river. Please notice that the purpose of this research is to explore the influence of river rectification and greening projects on the well-being index of people and cities.
Dear Reviewers:
Thanks for your suggestion. Description and discussion of hydraulic engineering. We add it in p.2. And join the discussion in the content of p.12-17.
- The title is not accurately describe the content of the manuscript. Where’s the study about environmental risk of epidemic?
Dear Reviewers:
Thanks for your suggestion. We have updated the title to include the topic of COVID-19.
Finally, we thank you again later.
Thank you for taking the time to provide valuable review suggestions.
We believe that this manuscript will be improved with revisions suggested by the review.
Thanks for your assistance.
Reviewer 2 Report
The article studies a very questionable subject and that is why it is very challenging as research.
First of all, some remarks need to be done about the writing of the paper. One of them is that the author should read once again the article because some parts of the text are too long and difficult to understand (a few examples are line 265-268, 337-338, 370-379). Another recommendation would be the care for the readers and their background. A paper with a subject like this would attract researchers from different fields and some of them would not understand very clear the statistic methods if they are not described as purpose, input data and meaning of the output. (For example, the KMO method is not explained at all, jut a value is presented and no reference about what that means, only a conclusion). In line 330 I think that is a mistake: it should be “above 0.6” instead of “above 0,06”.
I also must admit that the idea between lines 385-388 it is very well written.
Second, I think some remarks need to be done regarding the way that the subject is treated. The article is studying the human interventions that are made on both sides of a river, protecting the villages against floods states as the main purpose of the intervention. One of the main issues is that a quality analysis of the intervention is not provided by authors. The view of urban planners, architects, landscape architects and ecologists are not presented, the issue of sustainability for this project is not analyzed and the relation and effects within the existing context are not provided. The authors should provide details about the intervention, what was done to the river banks, what modifications were made to the public space, to the relation between public / private space and the water front. If there was a cognitive map that analyzed the memory of the place, the positive space that should be preserved as a landmark and the negative space that should have been the main area for change and reparatory actions. Those aspects are very important for understanding if the society changed with the intervention or if it is the same. There are possible scenarios that encourage people movement form one neighborhood to another, influenced by this type of actions and the questionnaires method could be compromised because it would analyze the perception of people who moved in this area just for those facilities.
Another possible issue regards a lack of comparison between this project and others that resemble this one, and the perception there. If another example is not provided as real scale of values would be missing and clear conclusions. The research is done from the statistic point of view and the needs of the community, needs of public space and public facilities are not analyzed and presented although they are a main element in happiness and well-being perception.
In the end I believe that the conclusions could be more specific and more concentrated on the outcomes (positive or negative) provided by this type of intervention and could be argued if this type of massive intervention is a good or bad thing for the society that lives there.
Author Response
Reviewer 2
The article studies a very questionable subject and that is why it is very challenging as research.
Dear Reviewers:
Thanks for your approval.
First of all, some remarks need to be done about the writing of the paper. One of them is that the author should read once again the article because some parts of the text are too long and difficult to understand (a few examples are line 265-268, 337-338, 370-379). Another recommendation would be the care for the readers and their background. A paper with a subject like this would attract researchers from different fields and some of them would not understand very clear the statistic methods if they are not described as purpose, input data and meaning of the output. (For example, the KMO method is not explained at all, jut a value is presented and no reference about what that means, only a conclusion). In line 330 I think that is a mistake: it should be “above 0.6” instead of “above 0,06”.
Dear Reviewers:
Thank you for your suggestion. We have fixed an issue with too much text in paragraphs within manuscripts. and grammar correction. And corrected the erroneous numbers. As in p.8.
Hope to get your approval.
I also must admit that the idea between lines 385-388 it is very well written.
Dear Reviewers:
Thanks for your approval.
Second, I think some remarks need to be done regarding the way that the subject is treated. The article is studying the human interventions that are made on both sides of a river, protecting the villages against floods states as the main purpose of the intervention. One of the main issues is that a quality analysis of the intervention is not provided by authors. The view of urban planners, architects, landscape architects and ecologists are not presented, the issue of sustainability for this project is not analyzed and the relation and effects within the existing context are not provided. The authors should provide details about the intervention, what was done to the river banks, what modifications were made to the public space, to the relation between public / private space and the water front. If there was a cognitive map that analyzed the memory of the place, the positive space that should be preserved as a landmark and the negative space that should have been the main area for change and reparatory actions. Those aspects are very important for understanding if the society changed with the intervention or if it is the same. There are possible scenarios that encourage people movement form one neighborhood to another, influenced by this type of actions and the questionnaires method could be compromised because it would analyze the perception of people who moved in this area just for those facilities.
Dear Reviewers:
Thank you for your suggestion. We supplement the professional statement of the relevant profession in our manuscript, and explain the representativeness of the interviewee.
Furthermore, a new discussion section is included based on the advice provided by various professionals. As shown in Table 1. and the above description in red font.
Finally updated all discussion narrative and meaning. As in p. 18-20.
Another possible issue regards a lack of comparison between this project and others that resemble this one, and the perception there. If another example is not provided as real scale of values would be missing and clear conclusions. The research is done from the statistic point of view and the needs of the community, needs of public space and public facilities are not analyzed and presented although they are a main element in happiness and well-being perception.
Dear Reviewers:
Thank you for your suggestion. Discussion on the inclusion of public construction and space. and added to the manuscript. As in p. 18-20.
In the end I believe that the conclusions could be more specific and more concentrated on the outcomes (positive or negative) provided by this type of intervention and could be argued if this type of massive intervention is a good or bad thing for the society that lives there.
Dear Reviewers:
Thank you for your suggestion. We have compiled new findings and updated abstracts based on new discussions. added to the manuscript. Such as 5.Conclusions and Abstract.
Finally, we thank you again later.
Thank you for taking the time to provide valuable review suggestions.
We believe that this manuscript will be improved with revisions suggested by the review.
Thanks for your assistance.
Reviewer 3 Report
The authors investigated the impact of river remediation and green space on urban happiness. They focused on environmental experience value, leisure involvement, satisfaction, and well-being. The authors examined the well-being index of people and cities. They found that, under current development, their focused elements were not consistent with one another. They reported that people had different views on various issues due to some factors. Their primary finding was that they could identify their interactions among them. The data is presented clearly. The reviewer agrees with the authors that the factor clarification brought by river remediation and green space would be of interest to readers.
Although this is an interesting study, there are some concerns in the present work.
- As there are many errors, the authors should carefully check the text. Please use an English editing service and check the English in this article. I cannot list all of them within the usual review period.
- In the abstract, all elements in this paper should be described. However, background information is missing, and it is necessary to add and describe it briefly.
- The content of “1. Introduction” is too long. 1.1 to 1.4 should move to “2. Methods.”
- “2.1 Research Process and Framework” describes the purpose of this study, which should move to “1. Introduction.”
- There are many too long sentences, P.2 L81-86, P.6 L257-260, P.10 L372-378, P.10 L378-384, P.11 L424-427, P.16 L22-28, P.16 L.34-P.17 L.39, P.17 L55-63, P.17 L63-70, P19 L.148-155, P.21 L244-248, P.21 L270-274, P.21 L278-282, P.22 L306-310, and P.22 L326-330, which should be separated into a few sentences to be read and understood easily.
- Obviously, this is minor, but some discriminative terms, such as mankind (->humankind), man-made (->artificial), and manpower (->workforce), should be revised.
- In the discussion, while this is an interesting and solid study, the data presented here seems to be merely correlative and does not show any causal relationship.
Author Response
Reviewer 3
The authors investigated the impact of river remediation and green space on urban happiness. They focused on environmental experience value, leisure involvement, satisfaction, and well-being. The authors examined the well-being index of people and cities. They found that, under current development, their focused elements were not consistent with one another. They reported that people had different views on various issues due to some factors. Their primary finding was that they could identify their interactions among them. The data is presented clearly. The reviewer agrees with the authors that the factor clarification brought by river remediation and green space would be of interest to readers.
Although this is an interesting study, there are some concerns in the present work.
Dear Reviewers:
Thanks for your approval.
As there are many errors, the authors should carefully check the text. Please use an English editing service and check the English in this article. I cannot list all of them within the usual review period.
Dear reviewers:
Thanks for your suggestion. The manuscript has been sent to English professionals to assist with proofreading. Such as the red text of the manuscript. Hope to have your approval.
In the abstract, all elements in this paper should be described. However, background information is missing, and it is necessary to add and describe it briefly.
Dear Reviewer
We appreciate your suggestion. As suggested, we have strengthened the background narrative and updated the manuscript. Such as 1. Introduction.
Hope to get your approval.
The content of “1. Introduction” is too long. 1.1 to 1.4 should move to “2. Methods.”
“2.1 Research Process and Framework” describes the purpose of this study, which should move to “1. Introduction.”
Dear Reviewer
We appreciate your suggestion. As suggested, we have repositioned two paragraphs and reworked the way the content is told. Looking forward to making the article more logical. Such as 1. Introduction and 3.1. Research Framework and Hypothesis.
Hope to get your approval.
There are many too long sentences, P.2 L81-86, P.6 L257-260, P.10 L372-378, P.10 L378-384, P.11 L424-427, P.16 L22-28, P.16 L.34-P.17 L.39, P.17 L55-63, P.17 L63-70, P19 L.148-155, P.21 L244-248, P.21 L270-274, P.21 L278-282, P.22 L306-310, and P.22 L326-330, which should be separated into a few sentences to be read and understood easily.
Dear Reviewer
We appreciate your suggestion. We have significantly revised paragraphs with long sentences, and re-summarized and reorganized the manuscript. Relevant fixes, presented in red text from p.2 onwards.
Hope to get your approval.
Obviously, this is minor, but some discriminative terms, such as mankind (->humankind), man-made (->artificial), and manpower (->workforce), should be revised.
Dear Reviewer
We appreciate your suggestion. We have adjusted the relevant wording. The content is rendered as blue font.
Hope to get your approval.
In the discussion, while this is an interesting and solid study, the data presented here seems to be merely correlative and does not show any causal relationship.
Dear Reviewer
We have compiled new findings and updated abstracts based on new discussions. added to the manuscript. Such as 5.Conclusions and 3.1. Research Framework and Hypothesis.
Finally, we thank you again later.
Thank you for taking the time to provide valuable review suggestions.
We believe that this manuscript will be improved with revisions suggested by the review.
Thanks for your assistance.
Reviewer 4 Report
Dear author,
please find the Comments and Suggestions for your study in the attachemnet.

Author Response
Reviewer 4
It is not recommended to publish the paper in its present form. There are several unclear concerns which should be settled prior to a publication. A major revision is suggested.
Dear Reviewer
We appreciate your suggestion. We have made the greatest efforts to significantly adjust the manuscript content.
Hope to get your approval.
The manuscript provides an interesting insight into the river remediation on the urban happiness index. This study has a scientific potential. However, it not appropriately described and developed. In the manuscript is missing the comprehensive climate characteristics of the study filed (temperature, precipitation, hydrological) characteristics).
Is the study area exploited and used for the whole year or only in the limited period of the year?
The detailed information about land use of the study area and the river basin should be provided. In addition, river remediation and green space measurements are not described, and it has to be included in this study.
Dear Reviewer
We appreciate your suggestion. We supplemented narratives on climate, water volume, hydrology, opening hours, etc. on the research topic. As in p.2.
Hope to get your approval.
The authors should also provide the different graphs for better results representation. Are the results of this study comparative from the relative studies? This should be clearly described in Discussion part.
Dear Reviewer
We appreciate your suggestion. We've updated the description images of the research topic's location, resources, and more. As shown in Figure 1.
Hope to get your approval.
The writing style is also not very attractive, the authors use expression e.g., scholars; I recommend reformulating the style. The river characteristics: hydrology, its morphology, relief are poorly described, and this information have to be added.
Dear Reviewer
We appreciate your suggestion. The researchers did their best to significantly adjust the content of the manuscript, hoping to present a clearer logic.
Hope to get your approval too.
Title: The name of the study is not chosen appropriately. The focus of the study is epidemic environmental risk. The title of the study should be reformulated.
Dear Reviewer
We appreciate your suggestion. We have updated the title to include the topic of COVID-19.
Abstract
The abstract is too general, the more detailed information should be incorporated. Line 18: The purpose is to explore the influence...What do you mean, purpose of what?
Line 20: It was found that under current development...
This is too general and vague. Development of what? Could you, please, provide the more consistent abstract, please? Summarize all the information from the whole study results and key findings to attract readers.
Keywords: These keywords are not related to this study, and they should be excluded from the manuscript: hydraulic engineering、landscape engineering;
Dear Reviewer
We compile new research findings, update abstracts and keywords based on new discussions. As shown in p.1.
Introduction
Line 41: According to their appearance and water volume, they can be divided into large and small rivers..
The division of the rivers is not correct from a hydrology point of view. You should specify river at the river basin level and the catchment. Well known and fixed hydrological definition should be used for its description.
Dear Reviewer
We appreciate your suggestion. We've adjusted the way the relevant text is told.
Line 45: The two types of rivers have had a significant influence on human survival and development.
Every river has an impact on human activity. I do not think that this definition is correct and accurate.
Dear Reviewer
We appreciate your suggestion. We've adjusted the way the relevant text is told.
Line 57: Hydraulic engineering...
This terminology is not properly used. Do you mean river renaturation?
Line 71: What is the development model? Please specify and describe!
Dear Reviewer
We appreciate your suggestion. We supplement the relevant narrative. As shown on p.2, "Water conservancy projects aim at....".
Line 73: Is this statement true? I really doubt that: The current policy decision is to reduce changes to river courses, avoid land reclamation, try to preserve the original appearance of the river...
Dear Reviewer
We appreciate your suggestion. We supplement the relevant narrative. As shown on p.2, "The current mainstream and trend of water conservancy policy [22-23] is to...".
Line 75: This is not flood control. I really wonder how by reducing changes to river courses can maintain the flood control. This must be revised!.
Dear Reviewer
We appreciate your suggestion. We supplement the relevant narrative. As shown on p.2, "The silt dug from the river course channel was dried, treated....".
Line 79: Mulan River...
The specific and detailed characteristics of the river has to be described in this section, e.g., how big is this catchment? What are its characteristics, land use....? Where is situated? Could you provide a map with the description? What about the detailed
characteristics of the study area?
Dear Reviewer
We appreciate your suggestion. We supplement the relevant narrative. As shown on p.2, "with an annual average temperature of 22.1°C and...". As shown in Figure 1.
Line 85: suffering heavy losses
This is too vague. What does it mean? Heavy losses of what?
Dear Reviewer
We appreciate your suggestion. We supplement the relevant narrative. As shown on p.2, "However, the Mulan River had been prone to floods since ancient times, ....".
Line 86 – 97: It is not at all clear from the text what kind of flood protection project it is.
Could you specify and explain it?
Dear Reviewer
We appreciate your suggestion. We supplement the relevant narrative. As shown on p.2, "The silt dug from the river course channel was dried, treated....".
Line 119 – 121: Furthermore, recently studies on river and hydraulic engineering have mainly aimed to the analyses of river flow velocity, water volume, water quality, flood prediction and ecological impact.
What were the results of these studies? It has to be in opposite way, e.g., the river flow velocity, water volume, water quality should be analyses for the flood prediction and ecological impact assessment.
Dear Reviewer
We appreciate your suggestion. We supplement the relevant narrative. As shown on p.3, "In response to climate change, some researchers ....".
Line 134: What is green space? It has to be specified.
Dear Reviewer
We appreciate your suggestion. We supplement the relevant narrative. As shown on p.4, "Green space along the riverbanks refers to the space of ....".
Line 132 – 140: In this section must be specified the study area. Is it whole Mulan River?
What is this study novelty and why this approach was chosen? This justification is neither clear nor innovative at all.
Dear Reviewer
We appreciate your suggestion. We supplement the relevant narrative. As shown in p.3, "In addition, recent studies on rivers and water conservancy ....". and p.9, 3.4. Scope and object description.
Line 260: What are the facilities generated by the Mulan River improvement and greening project. Could you, please describe them in more detail here?
Line 275: Figure 2 – Could you provide a detail description of this schema in this figure?
Dear Reviewer
We appreciate your suggestion. We supplement the relevant narrative. Such as p.2, "A total area of 5.5245 hectares of green ...." text description.
Line 328: The specification of the table is surely not correct. Do you mean university professor? There is name professor male – mentioned twice. This table is not comprehensive. Rewrite and better describe it! How many respondents for each category were asked? It has to be clear from the table, therefore modify it.
Dear Reviewer
We appreciate your suggestion. We have revised the presentation of Table 1 and added relevant instructions. As in Table 1.
Discussion
This study results should be compared with the similar studies and the main finding should be justified here. Could you, please describe and compare in detail?
Dear Reviewer
We appreciate your suggestion. Regarding the discussion part, the researcher has made significant revisions and adjusted the content of the discussion description. Such as 4.4. Discussion.
Conclusion
Identify the main outcomes on the base of the study results (describe the statistic values too).
Dear Reviewer
We appreciate your suggestion. The part about Conclusion. We have rearranged the narrative of the conclusion based on the updated discussion. Such as 5. Conclusions.
Finally, we thank you again later.
Thank you for taking the time to provide valuable review suggestions.
We believe that this manuscript will be improved with revisions suggested by the review.
Thanks for your assistanc
Round 2
Reviewer 2 Report
I think that the changes made to the article are good and it is an improvement but the ethical questions are still valid regarding the intervention and its users.
Author Response
Reviewer 2
I think that the changes made to the article are good and it is an improvement but the ethical questions are still valid regarding the intervention and its users.
Dear reviewers:
Thank you for your suggestion. We present the evidence presented in the literature 91, 93 studies. We considered that the subject of the study and the course of the investigation were free of drugs or invasive treatments. And all subjects participated in the experiment with informed and consent. This research process is a transparent, open, safe and fair process. There are no ethical issues involved. Such as 3.6 Ethical Considerations
Reviewer 3 Report
Thank you for revising the paper. I confirmed that this paper has been improved. I have the following minor concerns.
1) English errors in "Abstract"
P.1 L17 and 23 "project" and "environment" should be used in plural forms, projects, and environments, respectively.
P.1 L25 There is a typographical error. "during the" should be deleted.
P.1 L31 A blank after "knowledge" should be deleted.
P.1 L33 A definite article "the" should be added before "sustainable."
Please use an English editing service and check the English in this article. I cannot list all of them within the usual review period.
2) In the abstract, all elements in this paper should be briefly described. For example, as background information, "We have to conserve and protect river water resources as well as achieve sustainable development of them because they are essential for human life." should be added in the beginning.
3) 4) and 7) This article has been restructured and improved.
5) It is hard to read and understand too long sentences.
P.10 L406-408 This sentence is still long and can be improved, for example, be separated into a few sentences.
6) There is no discriminative term in this article.
Author Response
Reviewer 3
Thank you for revising the paper. I confirmed that this paper has been improved. I have the following minor concerns.
1) English errors in "Abstract"
Dear Reviewers:
Thank you for your suggestion. We have again adjusted the content and grammar of the abstract. Please refer to the abstract.
P.1 L17 and 23 "project" and "environment" should be used in plural forms, projects, and environments, respectively.
Dear Reviewers:
Thank you for your suggestion. We complete the correction of the relevant words.
P.1 L25 There is a typographical error. "during the" should be deleted.
Dear Reviewers:
Thank you for your suggestion. We complete the correction of the relevant words.
P.1 L31 A blank after "knowledge" should be deleted.
Dear Reviewers:
Thank you for your suggestion. We complete the correction of the relevant words.
P.1 L33 A definite article "the" should be added before "sustainable."
Dear Reviewers:
Thank you for your suggestion. We complete the correction of the relevant words.
Please use an English editing service and check the English in this article. I cannot list all of them within the usual review period.
Dear Reviewers:
Thank you for your suggestion. We have once again commissioned English professionals to assist in correcting the content of the manuscript.
2) In the abstract, all elements in this paper should be briefly described. For example, as background information, "We have to conserve and protect river water resources as well as achieve sustainable development of them because they are essential for human life." should be added in the beginning.
Dear Reviewers:
Thank you for your suggestion. The content of the summary has been adjusted again. Please refer to the abstract.
3) 4) and 7) This article has been restructured and improved.
Dear Reviewers:
Thank you for your suggestion. We have again adjusted the way the article is told.
5) It is hard to read and understand too long sentences.
P.10 L406-408 This sentence is still long and can be improved, for example, be separated into a few sentences.
Dear Reviewers:
Thank you for your suggestion. We have again adjusted the way the article is told. And modified the too long narrative. Such as L406-408.
6) There is no discriminative term in this article.
Dear Reviewers:
Thank you for your suggestion. We have again adjusted the way the article is told.

Reviewer 4 Report
Dear authors,
please find the suggestions and comments in the attachment of this e-mail.

Author Response
Reviewer 4
Title: The title of the study is not appropriate and suitable. It is not clear from the study that the green fields describe the situation from the COVID-19 Pandemic or green fields application in generally.
In addition, the title is too familiar and not suitable for research paper.
Dear Reviewers:
Thank you for your suggestion. We have fixed the title again.
Keywords: What does it mean? It is not comprehensive: “Environmental experience Value and Leisure Involvement.”
Dear Reviewers:
Thank you for your suggestion. We have re-selected appropriate nouns to be the keywords represented by the manuscript.
Lntroduction
Line 43: This sentence is not comprehensive, what do you mean? When water volume and flow velocity are stable and water hygiene and quality are stable.
Dear Reviewers:
Thank you for your suggestion. We have corrected the way the manuscript is narrated. L43.
Line 46: This is not a true statement, correct it.:
On the contrary, rivers will cause impact on the villages and their surrounding ecological environment, resulting in the risk of endangering life safety [6].
Dear Reviewers:
Thank you for your suggestion. We have corrected the way the manuscript is narrated.
Besides, the river influences the settlements in generally and not only the village...
Dear Reviewers:
Thank you for your suggestion. We have corrected the way the manuscript is narrated.
Line 50: I really doubt that: safeguard river water resources, plan a safe living environment for
people.
Dear Reviewers:
Thank you for your suggestion. We have corrected the way the manuscript is narrated.
This part of the text must be modified.
Dear Reviewers:
Thank you for your suggestion. We have corrected the way the manuscript is narrated.
Line 56: later studies
Could you provide more studies concerning to these subjects?
Dear Reviewer:
thanks for your advice. We update and supplement the relevant literature again.
Line 72: What does it means these characteristics: abundant water in the mountains.
Dear Reviewers:
Thank you for your suggestion. We have corrected the way the manuscript is narrated.
This is not characteristics for the regime of hydrology at all. Could you provide the detailed description of the hydrology balance for this study area?
Dear Reviewers:
Thank you for your suggestion. We have corrected the way the manuscript is narrated.
Line 85: Clearly describe and specify in detail: soft mattress
Dear Reviewer:
thanks for your advice. We supplement this term and update the relevant literature.
Line 90: Which disaster do you mean?
Dear Reviewer:
thanks for your advice. We have corrected the way the manuscript is narrated.
Figure 1: It does not provide the sufficient insight into the study area. Please provide the map the study area and map with the with the individual measures as well.
Dear Reviewer:
thanks for your advice. We have corrected the latest picture. As shown in Figure 1.
Line 99-121: This part is not comprehensive at all; it needs to be improved
Dear Reviewer:
thanks for your advice. We have corrected the way the manuscript is narrated.
What do you mean?: Scholars
Dear Reviewer:
thanks for your advice. We have corrected the way the manuscript is narrated.
This is not true: the researchers believe that the research gap can be filled by analyzing the influence of Mulan River...
Dear Reviewer:
thanks for your advice. We have corrected the way the manuscript is narrated.
Line 123-129: It is not conclusion, but introduction. Provide specification why did you hoose this study area and what is the benefit of this study. It is not clear from this part.
Dear Reviewer:
thanks for your advice. We have corrected the way the manuscript is narrated. L123-129
Why do you provide? Literature Review
Dear Reviewer:
thanks for your advice. We changed the name. In fact we are here analyzing the current state of the relevant research topic.
It is a scientific paper not literature review. You should mention the present state of the study here and compare it with the results of this study in conclusion part.
Dear Reviewer:
thanks for your advice.
We have changed the name and the current status and comparison of studies are mentioned in the manuscript.
Table 3 and 4 are illegible. They are not comprehensive, and it has to be modified.
Dear Reviewer:
thanks for your advice. We have revised Table 3 Table 4 again.
Table 5 and 6 are not described adequately in the text of the manuscript.
Dear Reviewer:
thanks for your advice. We supplement the descriptions in Table 5 and this part of Table 6.
In the part Discussion compare the results from this study with the other studies. 在Dear Reviewer:
thanks for your advice. We supplement the narrative in this section.
In addition, study area and improvement of the green fields is not described sufficiently. I would suggest adding more picture from the area in the study as well.
Dear Reviewer:
thanks for your advice.
We revise the inner narrative. And use Figure 1 to help readers understand the current construction of green space.
Happiness Index has to be described a explain its importance in this study. It is important.
Dear Reviewer:
thanks for your advice. The way our manuscript is narrated.
Could you, please, summarize the results in more comprehensive and consistent way? Dear Reviewer:
thanks for your advice. We've added a new chapter, narrative, and logical picture for a clearer summary. Such as 4.5 Recommended environmental development guidelines for urban river improvement and greening
